# Mitochondrial volume fraction and translation duration impact mitochondrial mRNA localization and protein synthesis

Tatsuhisa Tsuboi[1,2,3]*, Matheus P Viana[2†], Fan Xu[1], Jingwen Yu[1], Raghav Chanchani[1], Ximena G Arceo[1], Evelina Tutucci[4‡], Joonhyuk Choi[1], Yang S Chen[1], Robert H Singer[4,5,6,7], Susanne M Rafelski[2†]*, Brian M Zid[1]*

[1]Department of Chemistry and Biochemistry, University of California San Diego, La Jolla, United States; [2]Department of Developmental and Cell Biology and Center for Complex Biological Systems, University of California Irvine, Irvine, United States; [3]Division of Biological Science, Graduate School of Science, Nagoya University, Nagoya, Japan; [4]Department of Anatomy and Structural Biology, Albert Einstein College of Medicine, Bronx, United States; [5]Gruss-Lipper Biophotonics Center, Albert Einstein College of Medicine, Bronx, United States; [6]Department of Neuroscience, Albert Einstein College of Medicine, Bronx, United States; [7]Janelia Research Campus, Howard Hughes Medical Institute, Ashburn, United States

*For correspondence:
ttsuboi@ucsd.edu (TT);
susanner@alleninstitute.org (SMR);
zid@ucsd.edu (BMZ)

Present address: †Allen Institute for Cell Science, Seattle, United States; ‡Systems Biology Lab, Amsterdam Institute of Molecular and Life Sciences, Vrije Universiteit Amsterdam, Amsterdam, Netherlands

Competing interests: The authors declare that no competing interests exist.

**Abstract** Mitochondria are dynamic organelles that must precisely control their protein composition according to cellular energy demand. Although nuclear-encoded mRNAs can be localized to the mitochondrial surface, the importance of this localization is unclear. As yeast switch to respiratory metabolism, there is an increase in the fraction of the cytoplasm that is mitochondrial. Our data point to this change in mitochondrial volume fraction increasing the localization of certain nuclear-encoded mRNAs to the surface of the mitochondria. We show that mitochondrial mRNA localization is necessary and sufficient to increase protein production to levels required during respiratory growth. Furthermore, we find that ribosome stalling impacts mRNA sensitivity to mitochondrial volume fraction and counterintuitively leads to enhanced protein synthesis by increasing mRNA localization to mitochondria. This points to a mechanism by which cells are able to use translation elongation and the geometric constraints of the cell to fine-tune organelle-specific gene expression through mRNA localization.

## Introduction

Mitochondria are essential cellular organelles that are key sources of ATP generation via oxidative phosphorylation as well as the assembly of iron-sulfur clusters and many other catabolic and anabolic reactions (*Attardi and Schatz, 1988*). To support mitochondrial function, proteins from hundreds of nuclear genes are imported into mitochondria from the cytoplasm (*Morgenstern et al., 2017*). This has to be coordinated with the gene expression of the mitochondrial genome, which in *Saccharomyces cerevisiae* contains 13 genes (*Borst and Grivell, 1978*). While cells can generate ATP through mitochondrial oxidative phosphorylation, they can also use glycolysis as an alternative means of generating ATP. *S. cerevisiae* are Crabtree-positive yeast and will actively repress respiration and the use of alternative carbon sources in conditions in which the fermentable carbon source glucose is present (*De Deken, 1966*). This seems counterintuitive as the yield of ATP per glucose molecule is much higher in respiration compared to fermentation, but it is thought that fermentation allows higher fluxes of metabolite processing, leading to faster growth (*van Dijken et al., 1993*). Yet as

cells run out of glucose they must switch their primary ATP generation source from fermentation to respiration. This metabolic change is known to dramatically change the mitochondrial morphology (*Egner et al., 2002*). The protein content of yeast mitochondria also shows dynamic changes in response to shifting cellular energy demands (*Morgenstern et al., 2017*; *Paulo et al., 2016*). The HAP complex is known to play an important role in the transcriptional upregulation of mitochondrial biogenesis upon a shift to non-fermentable carbon sources (*Buschlen et al., 2003*). Translational regulation has also been found to be important in the control of mitochondrial gene expression as oxidative phosphorylation protein coding mRNAs gradually increase their protein synthesis as the growth environment changes from fermentative growth to respiratory conditions (*Couvillion et al., 2016*). mRNA localization is a means to post-transcriptionally regulate gene expression at both a temporal and spatial level (*Martin and Ephrussi, 2009*). In the 1970s, electron microscopy analysis found that cytoplasmic ribosomes can be localized along the mitochondrial outer membrane (*Kellems et al., 1974*). Recent microarray and RNA-seq analyses of biochemically fractionated mitochondrial membranes and fluorescent microscopy analysis have identified subsets of nuclear-encoded mRNAs that are mitochondrially localized (*Fazal et al., 2019*; *Gadir et al., 2011*; *Garcia et al., 2007*; *Marc et al., 2002*; *Saint-Georges et al., 2008*; *Williams et al., 2014*). It has been shown that both the 3′ UTR and coding regions, primarily through mitochondrial targeting sequences (MTSs), contribute to mitochondrial localization. One class of localized mRNAs (Class I) was shown to be dependent on the Puf3 RNA-binging protein through binding motifs in the 3′ UTR (*Saint-Georges et al., 2008*). Another class of mRNAs was localized to the mitochondria independently of Puf3 (Class II). Many of the localized mRNAs show reduced association upon polysome dissociation through EDTA or puromycin treatment, implicating translation as a necessary factor for mRNA localization (*Eliyahu et al., 2010*; *Fazal et al., 2019*). The mitochondrial translocase of the outer membrane (TOM) complex has been shown to impact mRNA localization through interaction with the nascent MTS (*Eliyahu et al., 2010*), while the outer-membrane protein OM14 has been shown to be a mitochondrial receptor for the ribosome nascent-chain-associated complex (NAC) (*Lesnik et al., 2014*). Isolating mitochondrially localized ribosomes to perform proximity-specific ribosome profiling revealed that many mitochondrial inner-membrane protein mRNAs are co-translationally targeted to the mitochondria (*Williams et al., 2014*). These observations have suggested a mechanism of co-translational protein import into mitochondria for a subset of nuclear-encoded mitochondrial mRNAs.

While mRNA localization is a way to control gene expression and there is strong evidence for the localization of mRNAs to the mitochondria, the use of this mRNA localization to alter the composition of mitochondria in different environmental conditions has not been explored. Furthermore, though mitochondrial biogenesis is thought to be transcriptionally regulated in relation to the metabolic needs of the cell, how these changing mitochondrial dynamics may also directly impact mRNA localization and protein synthesis has not been investigated. Here we report that interactions between mitochondria and mRNA/nascent-peptide (MTS) complexes can be altered by both the kinetics of protein synthesis and the fraction of cytoplasm that is mitochondrial, leading to condition-dependent mitochondrial mRNA localization during respiratory conditions. This localization subsequently leads to enhanced protein expression for these condition-specific localized mRNAs during respiratory conditions.

## Results

### mRNA association with mitochondria differs between fermentative and respiratory conditions

To explore how mRNA localization is impacted by the metabolic state of the cell and changing mitochondrial morphologies, we developed a methodology to quantify mRNA location and mitochondrial 3D structure in living cells. To do this, we visualized mitochondria using the matrix marker Su9-mCherry and single-molecule mRNAs with the MS2-MCP system every 3 seconds in a microfluidic device (*Figure 1A*; *Figure 1—figure supplement 1*). We reconstructed and analyzed the spatial relationship between the mRNAs and mitochondria using the ImageJ plugin Trackmate (*Tinevez et al., 2017*) and MitoGraph V2.0, which we previously developed to reconstruct 3D mitochondria based on matrix marker fluorescent protein intensity (*Rafelski et al., 2012*;

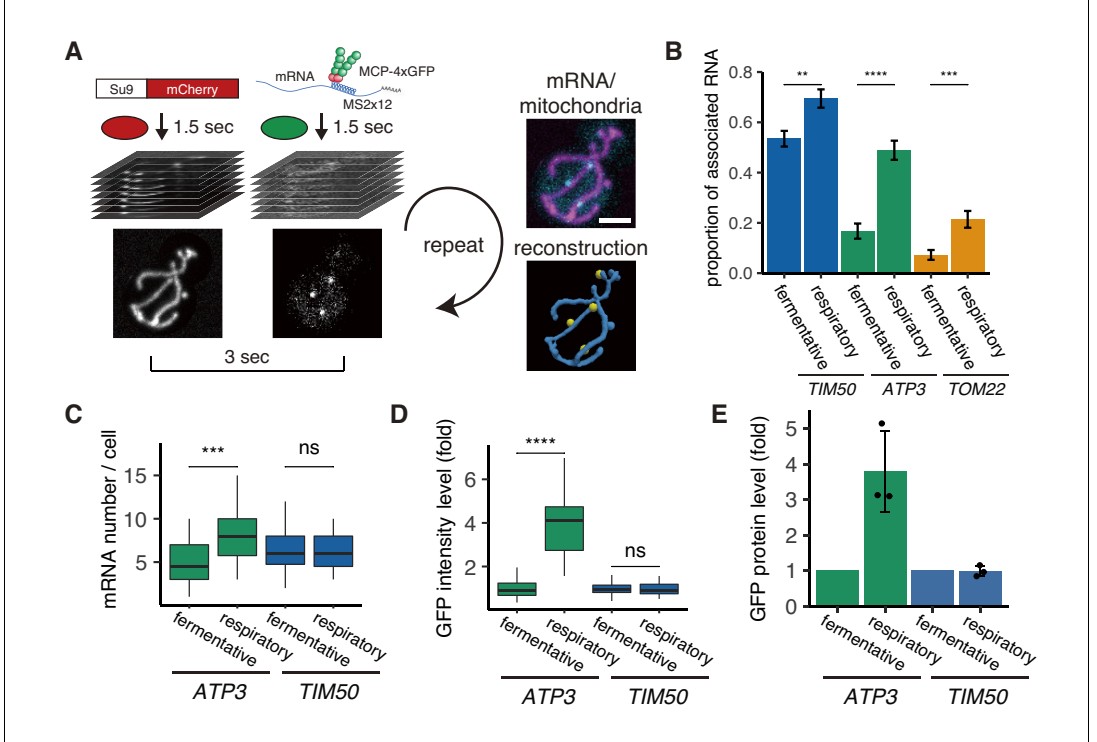

**Figure 1.** mRNA association with mitochondria differs between fermentative and respiratory conditions. (**A**) Experimental setup for live imaging. Mitochondria were visualized by Su9-mCherry and mRNAs were visualized by the single molecule MS2-MCP tethering system. Z-stacks were taken within 1.5 s for each individual channel, and multiple Z-stacks were merged into a series. (Right top) Z-projected image of a live cell. Cyan: mRNA, magenta: matrix. Scale bar, 2 μm. (Right bottom) Reconstructed mRNA and mitochondria. Yellow: mRNA, blue: mitochondria. (**B**) The proportion of mitochondrial associated mRNA per cell (n > 27) of the different mRNA species in fermentative and respiratory conditions. Error bar represents standard error of the mean (s.e.m.). (**C**) *ATP3* mRNA and *TIM50* mRNA expression number per cell. MCP-GFP foci were counted per cell (n > 27). (**D**) Atp3p-GFP and Tim50p-GFP fusion protein expression level using GFP fluorescent intensity per cell. (**C**)-(**D**) Statistical significance was assessed by Mann–Whitney U-test (****p<0.0001; ***p<0.001; **p<0.01; ns, no significant difference). (**E**) Atp3p-GFP and Tim50p-GFP fusion protein expression level using western blotting with anti-GFP antibody. Each dot represents an independent experiment. Error indicates standard deviation of three independent experiments.

The online version of this article includes the following figure supplement(s) for figure 1:

**Figure supplement 1.** Validation of live imaging mRNA foci by single molecule FISH.

**Figure supplement 2.** The distribution of mRNA-mitochondria distance in cycloheximide and 1,10-phenanthroline treated cells.

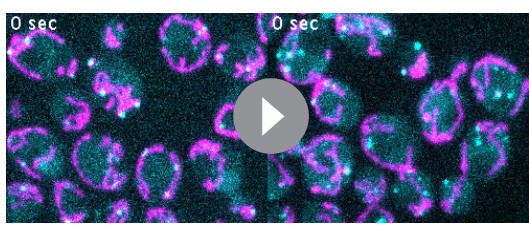

**Video 1.** Z-projection of live imaging of mRNA and mitochondria in cycloheximide and 1,10-phenanthroline treated cells. *TIM50* and *TOM22* mRNA were visualized 10 minuetes after the addition of cycloheximide and 1,10-phenanthroline in fermentative condition. (Left) *TIM50* mRNA (Right) *TOM22* mRNA. Cyan: mRNA, magenta: matrix.

https://elifesciences.org/articles/57814#video1

*Viana et al., 2015*; *Figure 1A*). We measured the distance between mRNA and mitochondria by finding the closest meshed surface area of the mitochondrial matrix (*Figure 1—figure supplement 2*; *Video 1*; Materials and methods).

We first analyzed three different mRNAs (*Saint-Georges et al., 2008*; *Williams et al., 2014*). Two mRNAs have previously been found to be mitochondrially localized and contain a mitochondrial targeting sequence (MTS): *ATP3* mRNA, which encodes the gamma subunit of ATP synthase, and *TIM50*, which encodes a component of the inner membrane translocase. The third mRNA, *TOM22*, encodes an outer-membrane translocase that does not contain an MTS and has previously been found to be predominantly diffusely

localized (*Gadir et al., 2011*; *Garcia et al., 2010*; *Williams et al., 2014*). During fermentative growth, we observed *TIM50* mRNA to be strongly associated with the mitochondria, whereas *TOM22* showed low association consistent with previous studies (*Figure 1B*; *Video 2*; Materials and methods). Even though *ATP3* had previously been categorized as a mitochondrially localized mRNA (*Gadir et al., 2011*; *Saint-Georges et al., 2008*), we unexpectedly found this to be condition-dependent as it has low association with mitochondria, similar to *TOM22*, in fermentative conditions. However, during respiratory conditions it strongly shifted towards association with the mitochondrial surface, in a manner more similar to *TIM50* (*Figure 1B*). This means that nuclear-encoded mitochondrial mRNAs do not have to be solely mitochondrially localized or diffusely localized; instead, they can show a switch-like behavior depending on the metabolic state of the cell.

There are large changes in mitochondrial composition as yeast switch from fermentative to respiratory metabolism (*Morgenstern et al., 2017*). To investigate whether there may be a relationship between mRNA localization and gene expression, we measured mRNA levels via number of mRNAs per single cell and protein levels via both single-cell measurements and bulk assays in both fermentative and respiratory conditions. For the condition-dependent *ATP3* mRNA, we found that protein levels increased 4-fold, whereas mRNA levels increased less than 2-fold when cells were grown in respiratory versus fermentative conditions. *TIM50* mRNA, which is constitutively localized to the mitochondria under both conditions, showed no change in protein or mRNA levels in respiratory conditions (*Figure 1C–E*). This suggests a relationship between mRNA localization to the mitochondria and protein production.

## Relationship of mRNA localization to mitochondrial volume fraction

As yeast cells shift to respiratory conditions, mitochondrial biogenesis increases the mitochondrial volume while the cell cytoplasmic volume decreases, thus leading to an increase in the mitochondrial volume fraction in respiratory conditions (*Egner et al., 2002*; *Figure 2A*; *Figure 2—figure supplement 1*). While *ATP3* mRNA showed a strong condition-dependent localization, *TIM50* and *TOM22* mRNAs also showed modestly increased mitochondrial association during respiratory conditions (*Figure 1B*). We wondered what impact the reduction in the availability of free cytoplasmic space due to mitochondrial expansion had on mRNA co-localization, especially for *TOM22*, which is not known to bind to the mitochondria. To test this, we quantified both the mitochondrial localization of each mRNA and changes in mitochondrial volume fraction at a single-cell level. We found that *TOM22* showed a linear increase in co-localization that was directly proportional to mitochondrial volume fraction (*Figure 2B*). We also found that *ATP3* mRNA was more sensitive to mitochondrial volume fraction than *TIM50* and *TOM22*. This sensitivity was independent of nutrients, as fermentative yeast cells showed a larger increase in *ATP3* localization as the mitochondrial volume fraction increased (*Figure 2B*). At the lowest mitochondrial volume fractions, *ATP3* localization was similar to the diffusely localized *TOM22* mRNA, whereas at the highest volume fractions, its localization was close to the mitochondrially localized mRNA *TIM50*. This suggests that increased mitochondrial volume fraction in turn increases *ATP3* mRNA localization to mitochondria.

To further test our hypothesis that mRNA localization is regulated by mitochondrial volume fraction, we designed *in silico* experiments based on our experimentally measured cell and mitochondrial boundaries and used a mathematical model to investigate how particles of varying affinities would co-localize with mitochondria (*Figure 2C*). We were able to recapitulate the behavior of *TOM22* via a model of an ideal Brownian particle with no affinity for the mitochondria, which showed linearly correlated localization with mitochondrial volume fraction (*Figure 2C, equation (1)*; *Figure 2D*; *Figure 2—figure supplement 2A*). We then set up a simple equilibrium equation where the baseline equilibrium constant, $K_0$, was set by a freely diffusing particle like *TOM22* and multiplied by the affinity, A, of the particle for the mitochondria, thus giving $K = AK_0$ (*Figure 2C, equation (2)*; Materials and methods). As the value of A

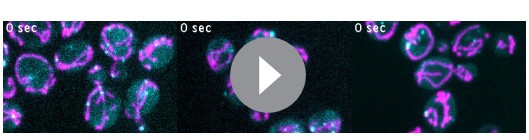

**Video 2.** Z-projection of live imaging of mRNA and mitochondria in fermentative condition. *TIM50, ATP3* and *TOM22* mRNA were visualized in fermentative condition. (Top) TIM50 mRNA (Middle) ATP3 mRNA (Bottom) TOM22 mRNA. Cyan: mRNA, magenta: matrix.
https://elifesciences.org/articles/57814#video2

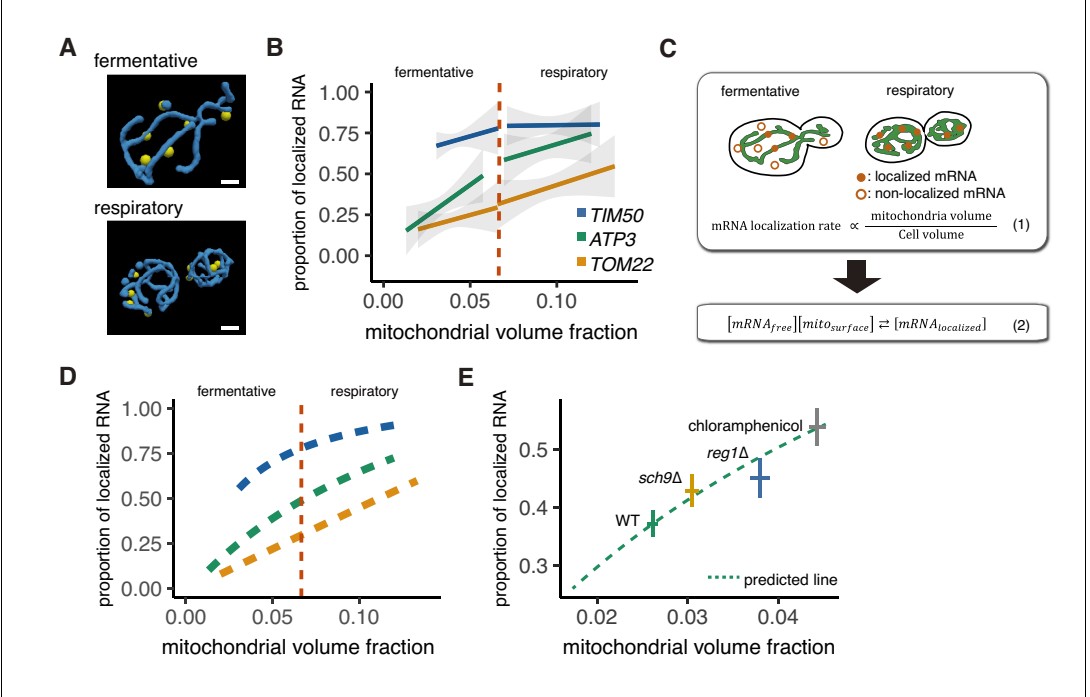

**Figure 2.** Mitochondrial volume fraction correlates with mRNA localization. (**A**) Snapshot of reconstructed mitochondrial surface (blue) and *TIM50* mRNA foci (yellow) in fermentative and respiratory conditions. Scale bar, 1 µm. (**B**) Relationship between the mitochondrial volume fraction and the proportion of mRNA localization to mitochondria. Trend line was depicted according to the best linear fit of the proportion of localization and mitochondrial volume fraction of single cells in each condition of different mRNAs (n > 27). Dotted red line marks the difference between fermentative and respiratory conditions for mitochondria volume fraction. Gray region surrounding the trend lines represents the 95% confidence interval (CI) for each line. (**C**) Schematic of *in silico* experiment. (Top) Brownian particle distribution indicates that mRNA localization rate correlates with mitochondrial volume fraction. (Bottom) Hypothetical thermodynamic equilibrium of binding of mRNA to mitochondria. [mRNA$_{localized}$], mitochondrial localized mRNA; [mRNA$_{free}$], free diffusing mRNA; [mito$_{surface}$], mitochondrial surface where mRNA can bind. (**D**) Mitochondrial volume fraction and mRNA localization have stoichiometric correlation. Relationship of proportion of mRNA localization to mitochondria and mitochondrial volume fraction from mathematical modeling was plotted. Yellow line represents linearly fitted line for *TOM22* mRNA. Green and blue lines were plotted through equilibrium constant of 2.4$K_0$ and 8.8$K_0$, respectively, as described in Materials and methods. (**E**) In glucose conditions, *sch9Δ* and *reg1Δ* mutant strains as well as chloramphenicol addition exhibit higher mitochondrial volume fraction and increase the localization of *ATP3* mRNA to the mitochondria (n > 27). Predicted relationship of proportion of mRNA localization to mitochondria and mitochondrial volume fraction from mathematical modeling of WT strains was plotted as a green dotted line. Mean values of each axis for WT, mutant cells, and cells with chloramphenicol addition were plotted as crosses. Error bar represents s.e.m.

The online version of this article includes the following figure supplement(s) for figure 2:

**Figure supplement 1.** Higher mitochondrial volume fraction is a feature of respiratory conditions.

**Figure supplement 2.** Mitochondrial volume fraction and mRNA localization have stoichiometric correlation.

**Figure supplement 3.** Mitochondrial volume fraction correlates with *ATP3* mRNA localization in WT, *sch9Δ* and *reg1Δ* mutant strains, and upon chloramphenicol addition.

**Figure supplement 4.** *reg1Δ* mutant strains and chloramphenicol addition increases mitochondrial volume fraction.

**Figure supplement 5.** *OM14* and *OM45* transcripts level in fermentative and respiratory conditions.

increased in the simulation, the proportion of mitochondrial localization of the mRNA for a given mitochondrial volume fraction increased as well (*Figure 2—figure supplement 2B*). We then applied this relationship to estimate the experimental values of A to be 2.4 and 8.8 for *ATP3* and *TIM50*, respectively (*Figure 2D*; *Figure 2—figure supplement 2C*). Notably, mitochondrial volume fraction correlates with mRNA localization, but cell volume and mitochondrial volume alone did not show any correlation through fermentative and respiratory conditions (*Figure 2—figure supplement 2C–E*). Interestingly, this simple mathematical relationship also recapitulates the shape of the curves in *Figure 2B*, suggesting that the apparent shift in association that occurs during the switch from fermentative to respiratory conditions may be primarily a result of the combination of the mitochondrial

volume fraction and the strength of mRNA-specific association and, therefore, not due to the difference in growth condition or to other mechanisms.

To further test this model, we sought to manipulate mitochondrial volume fraction in a multitude of ways. We first tested the impact of increased mitochondrial biogenesis on mRNA localization while maintaining the nutrient conditions of the cell. Reg1 is a protein necessary for glucose repression in yeast, and *reg1Δ* cells exhibit increased mitochondrial function in glucose media (*Adachi et al., 2017*; *Hübscher et al., 2016*). Supporting our model, we observed increased mitochondrial volume fraction in rich glucose conditions in *reg1Δ* mutant cells as well as increased *ATP3* mRNA localization to the mitochondria (*Figure 2E*; *Figure 2—figure supplements 3A–D* and *4*). Additionally, the antibiotic chloramphenicol is known to decrease mitochondrial protein synthesis while having no effect on cytoplasmic translation and has been found to increase the fraction of mitochondrial protein in *Neurospora crassa* and mitochondrial volume in *Tetrahymena pyriformis* (*Brody, 1992*; *Gleason et al., 1975*). We sought to test whether chloramphenicol would alter the mitochondria volume fraction in yeast during rich glucose conditions when mitochondrial respiration is not required. We observed an ~70% increase in mitochondria volume fraction in yeast cells treated with chloramphenicol and, simultaneously, a ~ 50% increase in *ATP3* mRNA mitochondrial association during vegetative conditions (*Figure 2E*; *Figure 2—figure supplements 3A–D* and *4*). These results support the hypothesis that mitochondrial volume fraction is the important variable for *ATP3* mRNA localization and not a factor related to respiration, as chloramphenicol decreases respiratory function (*Williamson et al., 1971*) while increasing mitochondrial volume fraction and *ATP3* localization.

Finally, as mitochondrial volume fraction can be increased by either increasing mitochondrial volume or decreasing cytoplasmic volume, we tested whether decreasing cytoplasmic volume could increase *ATP3* mRNA mitochondrial association. *sch9Δ* was previously found to be one of the smallest strains from a genome-wide screen for cell size (*Jorgensen et al., 2002*). We found that *ATP3* mRNAs showed increased mitochondrial association in *sch9Δ*, though from initial analysis this was not accompanied by an increase in mitochondrial volume fraction. While imaging these cells, we noticed a large increase in the relative vacuole in *sch9Δ* cells; the vacuole volume fraction was ~6 x higher in *sch9Δ* cells versus WT cells. As the vacuole restricts the accessible cytoplasmic volume, we recalculated the mitochondrial volume fraction relative to the accessible cytosol and found that *sch9Δ* cells have a significant increase in mitochondrial volume fraction after this correction (*Figure 2—figure supplements 3E–F* and *4*). With this corrected mitochondrial volume fraction, we found that, similar to *reg1Δ* and chloramphenicol-treated cells, *sch9Δ* cells had increased localization of *ATP3* mRNA to the mitochondria that was not significantly different either from our experimental measurements relating mitochondrial volume fraction and mRNA association in WT cells or from what would be predicted from our mRNA localization simulation based solely on changing mitochondrial volume fraction (*Figure 2E*; *Figure 2—figure supplements 3G–I* and *4*). These results show that the relationship between mitochondrial volume fraction and mRNA localization holds across highly varied perturbations, including nutritional (glucose/glycerol), genetic (*reg1Δ*, *sch9Δ*), and pharmacological (chloramphenicol).

An alternative possibility is that expression of the import machinery or the NAC receptor correlate with mitochondrial volume fraction and that these are what is important for the changes in *ATP3* mRNA localization. If this is the case, we expect that the expression of these genes should increase across all perturbations that have higher mitochondrial volume fraction and increased *ATP3* localization. To test this, we investigated the expression of potential known receptors in fermentative and respiratory conditions using published proteomics data (*Morgenstern et al., 2017*). Surprisingly, most of these potential receptors, including the import machinery, TOM/SAM complex, do not show a difference in expression between respiratory and fermentative conditions (*Figure 5—source data 1*). The outer membrane receptors OM14 and OM45 did have a significant 7 to 9-fold increase in expression when mitochondrial volume fraction is expanded. To further explore the role of these outer membrane proteins in mRNA localization we measured their expression in *reg1Δ* and *sch9Δ* mutant strains, which have increased mitochondrial volume fraction and *ATP3* localization, but are grown in glucose media (*Figure 2E*). While we are able to recapitulate the large increase in expression of *OM14* and *OM45* in respiratory conditions, we do not find a significant increase in the expression of these RNAs in *reg1Δ* and *sch9Δ* mutant strains (*Figure 2—figure supplement 5*). Overall, these data point to the importance of increasing mitochondrial volume fraction effectively

shrinking the cytoplasm and favoring random interactions with the mitochondrial membrane, which would result in altered association based on binding affinities of the components involved. Furthermore, mRNA association with mitochondria can be tuneable to permit a switch-like transition in mitochondrial localization, as is seen for *ATP3* mRNA (*Figure 1B*), due to a nutrient-induced change in mitochondrial volume fraction.

## Translation regulates mRNA localization

Given these results, we wanted to delve further into the mechanism of this varying localization and protein production. Even though the mitochondrial protein import machinery is well described, an ER-like signal recognition particle dependent mechanism of co-translational protein import has not been identified for the mitochondria (*Golani-Armon and Arava, 2016*; *Reid and Nicchitta, 2015*). However, a series of biochemistry and microscopy analysis showed that some nuclear-encoded mitochondrial protein mRNAs are translated on the mitochondrial surface (*Gadir et al., 2011*; *Garcia et al., 2007*; *Gold et al., 2017*; *Kellems et al., 1974*; *Marc et al., 2002*; *Saint-Georges et al., 2008*; *Williams et al., 2014*). We therefore investigated the effects of the MTS and of protein translation on mRNA association with mitochondria. We remove or replaced the MTS of Tim50p with an ER-localization signal or introduced an ER-targeting signal at the N-terminus of Tom22p (*Wu et al., 2016*). Even though *TIM50* mRNA was associated with mitochondria, *ER-TIM50*, *TOM22*, *ER-TOM22,* and *TIM50-ΔMTS* mRNAs were not associated with mitochondria (*Figure 3A*, *Figure 3—figure supplement 1*), indicating that the *TIM50* MTS is necessary to recruit this mRNA to mitochondria. To further support the role of the MTS in mRNA localization, we tested whether reducing ribosome-nascent chain association by using the translation initiation inhibitor lactimidomycin (LTM) would affect mRNA localization. We found that *TIM50* mRNA in all conditions and *ATP3* mRNA in respiratory conditions decreased localization to the mitochondrial surface upon LTM addition, while *ATP3* mRNA in fermentative conditions showed only minimal changes in localization upon LTM addition (*Figure 3A*). These results suggest that the actively translating ribosome drives mRNA localization to mitochondria through production of the nascent N-terminal MTS.

From our simulation, we were able to recapitulate our experimental observation that *TIM50* mRNA has higher affinity for the mitochondria than *ATP3* mRNA, making its localization less dependent on mitochondrial volume fraction. As the MTS was necessary for localization to the mitochondria, our initial hypothesis was that Tim50p MTS has a higher affinity for the mitochondria than the Atp3p MTS, causing the differences in mRNA affinities. To test this hypothesis, we designed chimeric reporter genes wherein we swapped the MTS sequences between Tim50p and Atp3p under *TIM50* promoter control (*Figure 3B*). Surprisingly, we found that the downstream coding sequence (CDS) was what differentiated *TIM50* from *ATP3*, not the MTS. When the reporter gene contained the TIM50-CDS, it showed uniform protein production in fermentative versus respiratory conditions, independent of which MTS was present (*Figure 3C*, lane 1 vs. 2 and lane 5 vs. 6). However, when the reporter gene contained the ATP3-CDS, it showed decreased protein production in fermentative conditions (*Figure 3C*, lane 3 vs. 4 and lane 7 vs. 8). Similarly, the reporter genes that harbored the ATP3-CDS also showed decreased proportion of mitochondrial associated mRNA in fermentative but not in respiratory conditions (*Figure 3D*, *Figure 3—figure supplement 2*). These experiments suggest that the *TIM50* and *ATP3* MTS have similar affinities for the mitochondria and, importantly, what drives the condition-specific differences in mRNA localization and protein production between these mRNAs is encoded in the downstream CDS.

## Ribosome stalling is important for constitutive mitochondrial localization

Our model proposes that the reason *ATP3* mRNA increases localization in respiratory conditions is that the increased mitochondrial volume fraction increases the probability that the nascent MTS will interact with the mitochondrial surface. If the *ATP3* and *TIM50* MTS have similar affinity for the mitochondria, we hypothesized that the reason *TIM50* has higher mitochondrial association at lower mitochondrial volume fractions is because the downstream CDS increases the chance of association between mitochondria and MTS, possibly by slowed translation elongation. Upon further examination, we found that the *TIM50* CDS has a run of 7 consecutive proline codons approximately 60 amino acids downstream of the MTS. Polyproline stretches have been shown to mediate ribosome

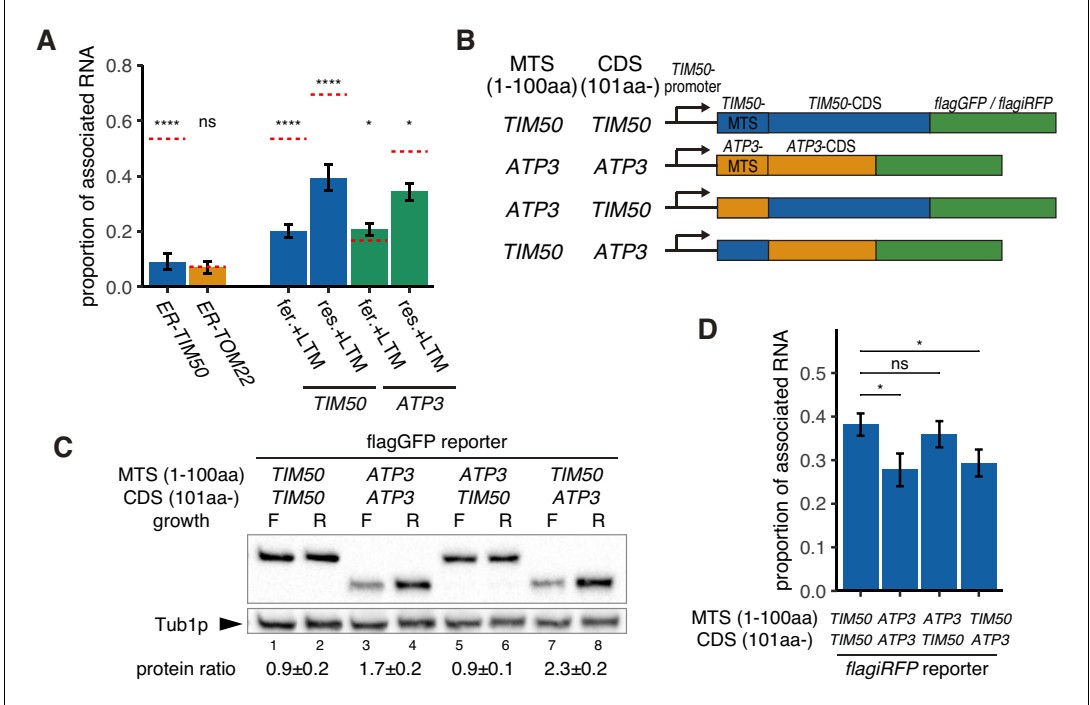

**Figure 3.** Increased protein synthesis and mRNA localization is regulated by the downstream coding sequence. (A) An ER-localization signal and translation inhibitor drugs alter the proportion of mitochondrial associated mRNA per cell of the strains in *Figure 1B* (n > 20). LTM indicates 50 mM lactimidomycin for 20 min. Error bar represents s.e.m. Statistical significance compared with control (value in *Figure 1B*, red dotted line) was assessed by Mann–Whitney U-test (****p<0.0001; *p<0.05; ns, no significant difference). (B) Schematic of chimeric reporter genes for swapping of MTS (1-100aa) and CDS (101aa-) between *TIM50* and *ATP3*. Chimeric genes were conjugated with either *flagGFP* or *flagiRFP*. (C) Protein expression from reporter genes with flagGFP depicted in (B). Growth 'F' represents fermentative and 'R' represents respiratory conditions. Tub1p was used as internal loading control. Protein expression ratio between respiratory and fermentative conditions is shown in the bottom row. Error indicates standard deviation of three independent experiments. (D) The proportion of mitochondrial associated mRNA per cell of reporter mRNAs *with flagiRFP* in fermentative conditions (n > 34). Error bar represents s.e.m. Statistical significance was assessed by Mann–Whitney U-test (*p<0.05; ns, no significant difference).

The online version of this article includes the following figure supplement(s) for figure 3:

**Figure supplement 1.** Mitochondrial targeting signal is required for *TIM50* mRNA localization to mitochondria.

**Figure supplement 2.** The reporter mRNAs show similar association with mitochondria in respiratory conditions.

stalling, and, when we investigated a ribosome profiling data set, we found that ribosomes accumulate at this polyproline stretch during fermentative conditions (*Zid and O'Shea, 2014*; *Figure 4—figure supplement 1*). This suggests a possible mechanism, similar to what has been seen for SRP recognition, by which local slowdown of ribosomes increases the chance that the mitochondria will recognize the *TIM50* MTS and consequently promote its association with the mitochondrial surface (*Pechmann et al., 2014*; *Zhang and Shan, 2012*). To test this, we deleted these polyproline residues and found this caused *TIM50* to be more sensitive to environmental conditions as it reduced both protein synthesis and mRNA localization of *TIM50* during fermentative conditions (*Figure 4A–D*). In contrast, the ATP3 coding sequence does not have any obvious strong ribosome stalling sequence. This suggests that *ATP3* mRNA localization and protein synthesis are regulated solely in a manner dependent on mitochondrial volume fraction. If this is true, artificially slowing the ribosomes in fermentative conditions should drive *ATP3* mRNA to become mitochondrially localized. To test this hypothesis, we inserted a run of 7 consecutive proline codons at 100 amino acids downstream of the start codon and analyzed protein expression in fermentative conditions (*Figure 4A*, right). We observed that protein production is increased 1.6-fold (*Figure 4E*) and mRNA localization is increased as well (*Figure 4F*). An alternative explanation is that the RNA sequence encoding the polyprolines is important for localization to the mitochondria and not ribosome stalling due to the encoded polyprolines. We believe this is not the case as translation initiation is key for the localization of *TIM50* mRNA to the mitochondria (*Figure 3A*), and that upon lactimidomycin treatment

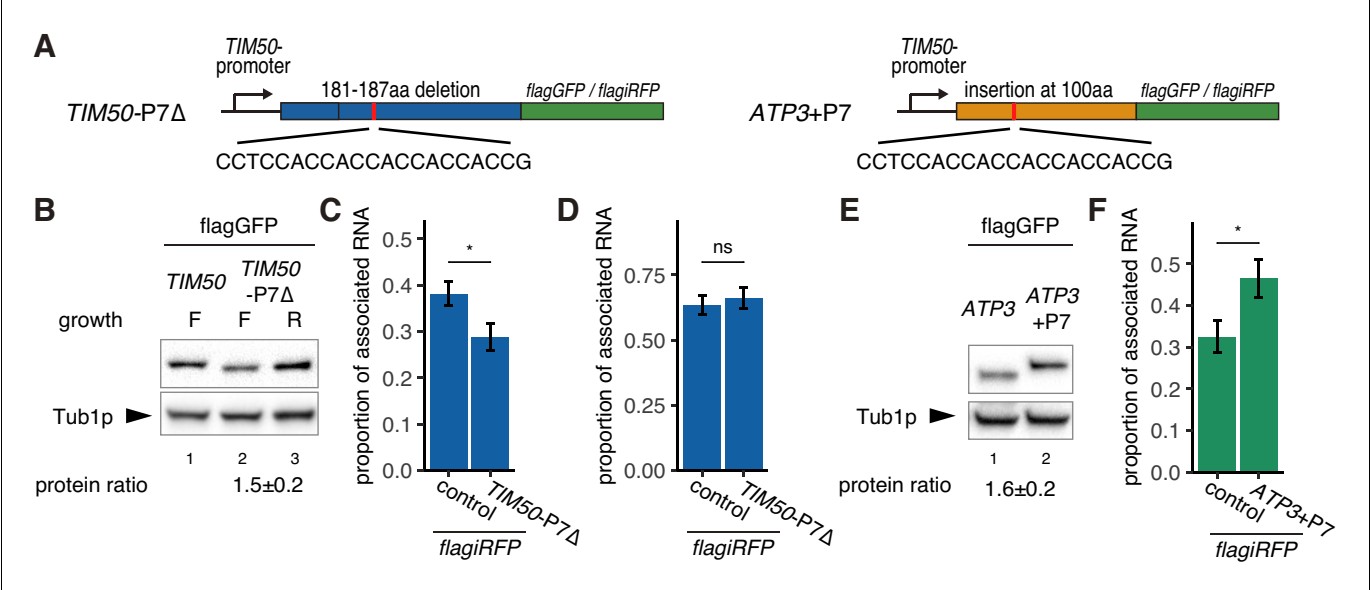

**Figure 4.** Decreased translational elongation localizes mRNA to mitochondria. (**A**) Schematic of deletion of polyproline sequence from *TIM50-flagGFP/flagiRFP* reporter gene and insertion of polyproline sequence into *ATP3- flagGFP/flagiRFP* reporter gene. These constructs are called *TIM50-P7Δ* and *ATP3+P7*. (**B**) Protein expression from reporter genes *TIM50-flagGFP* and *TIM50-P7Δ-flagGFP*. Growth 'F' and 'R' correspond to fermentative and respiratory conditions, respectively. Tub1p was used as internal loading control. Protein expression ratio between respiratory and fermentative conditions is shown in the bottom row. Error indicates standard deviation of three independent experiments. (**C, D**) The proportion of the mitochondrial associated mRNA per cell (n > 20) of the reporter mRNAs *TIM50-flagiRFP* and *TIM50-P7Δ-flagiRFP* in fermentative (**C**) and respiratory (**D**) conditions. (**E**) Protein expression from reporter genes *ATP3-flagGFP* and *ATP3+P7-flagGFP* in respiratory condition. Tub1p was used as internal loading control. Protein expression ratio between the reporter genes is shown in the bottom row. Error indicates standard deviation of three independent experiments. (**F**) The proportion of mitochondrial associated mRNA per cell (n > 29) of the reporter mRNAs *ATP3-flagiRFP* and *ATP3+P7Δ-flagiRFP* in fermentative conditions.

The online version of this article includes the following figure supplement(s) for figure 4:

**Figure supplement 1.** Ribosomes are enriched at a polyproline sequence of *TIM50* mRNA in a previously generated ribosome profiling dataset.

*TIM50* and *ATP3* have indistinguishable localization phenotypes even though *TIM50* mRNAs contain the RNA sequence for seven consecutive polyprolines and *ATP3* does not, arguing against the RNA sequence directly having a role in the differential localization between *TIM50* and *ATP3*. Overall these results further suggest that ribosome stalling leads to mRNA localization to mitochondria and increased protein production.

To further explore whether slowing translation elongation stabilizes the mRNA-ribosome complex with the MTS, thereby giving it more time to associate with mitochondria, we measured mitochondrial mRNA localization following the addition of the translation elongation inhibitor cycloheximide (CHX) (*Figure 5A*). As our hypothesis predicted, we observed a 3-fold increase in the association of *ATP3* mRNA with mitochondria during fermentative conditions but no increase in the case of *TOM22* mRNA (*Figure 5A*). Interestingly, CHX treatment only slightly increased *TIM50* mRNA localization. This potentially suggests that a portion of these mRNAs are incompetent for binding, perhaps due to nuclear localization or being in the midst of degradation, and that the majority of competent *TIM50* mRNAs were already localized to the mitochondria.

A previous study found that 130 of 551 annotated nuclear-encoded mitochondrial mRNAs are sensitive to translation elongation rate and become localized to the mitochondrial surface upon cycloheximide treatment, similar to *ATP3* (*Figure 5B*; *Williams et al., 2014*). Interestingly, all of the ATP synthase subunits that are conserved from bacteria to eukaryotes are sensitive to cycloheximide, except for the ε subunit, *ATP16*, whereas all of the nonconserved subunits are insensitive to cycloheximide (*Table 1*). We wondered whether this sensitivity may be indicative of mRNAs that also switch their localization as the mitochondrial volume fraction increases during respiratory conditions (*Table 1*). We tested *ATP2*, the conserved β subunit of the ATP synthase, and found that it was

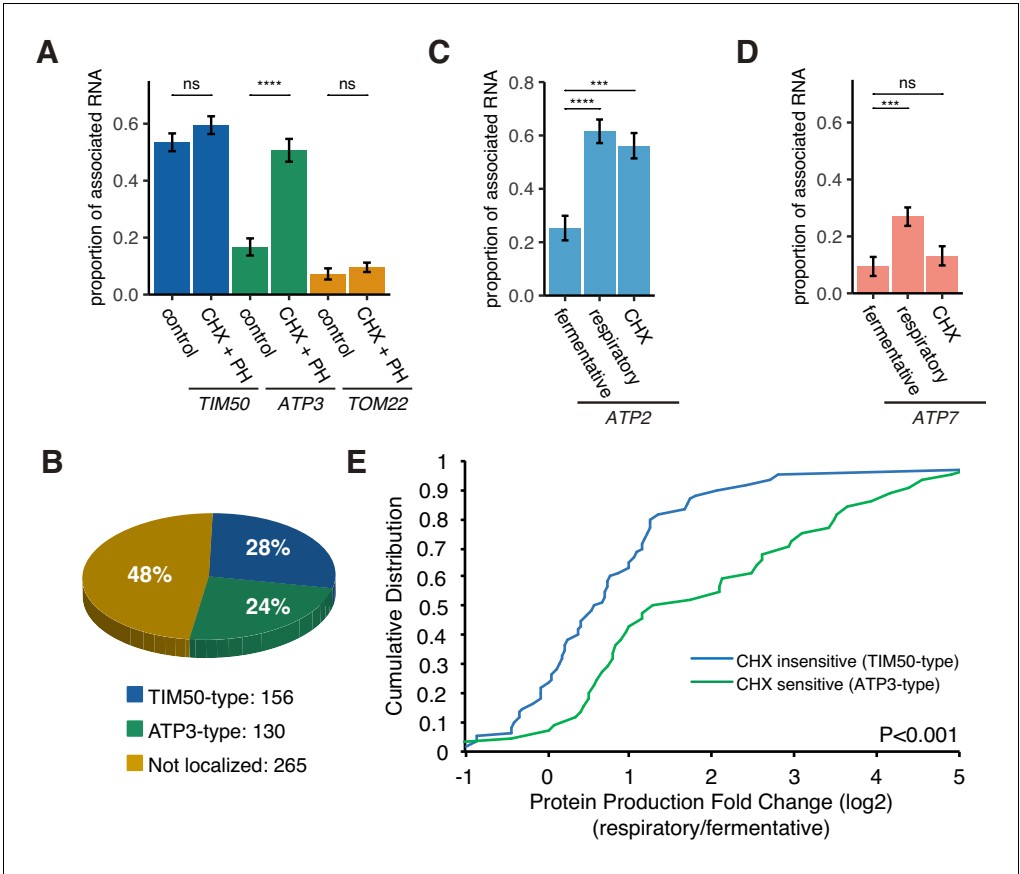

**Figure 5.** Decreased translational elongation localizes mRNA to mitochondria. (**A**) Translational inhibitor drugs alter the proportion of mitochondrial-associated mRNA per cell (n > 43) of the strains in *Figure 1C*. CHX+PH indicates 100 μg/mL cycloheximide and 200 μg/mL 1,10-Phenanthroline for 10 min in fermentative condition. Error bar represents s.e.m. (**B**) Percent distribution of *TIM50* and *ATP3* mRNA mitochondrial localization behavior in annotated mitochondria protein encoding genes. We re-summarized and classified TIM50-type mRNA as enriched at the mitochondria regardless of CHX addition and ATP3-type mRNA as enriched at the mitochondria only upon CHX addition using proximity ribosome profiling data of annotated mitochondria protein encoding genes (Mitop2) (*Elstner et al., 2009*; *Williams et al., 2014*). All other mRNAs were considered not localized. (**C**)-(**D**) The proportion of the mitochondrial associated mRNA per cell (n > 16) of MS2-tagged endogenous *ATP2* and *ATP7* mRNAs in fermentative, respiratory, and CHX-treated for 10 min in fermentative conditions. Error bar represents s. e.m. Statistical significance in this figure was assessed by Mann–Whitney U-test (****p<0.0001; ***p<0.001; **p<0.01; *p<0.05; ns, no significant difference). (**E**) Protein production is increased upon a switch from fermentative to respiratory conditions in ATP3-type genes. Cumulative distribution of protein production (ribosome reads) (*Couvillion et al., 2016*) of TIM50-type (n = 60) and ATP3-type genes (n = 44) were depicted with blue and green lines, respectively. Statistical significance in this figure was assessed by Student's t-test. Source data of ribosome profiling used for the analysis are summarized in the Source data 1.

The online version of this article includes the following source data and figure supplement(s) for figure 5:

**Source data 1.** Summary of transcriptomics and proteomics from previously published data.

**Figure supplement 1.** Mitochondrial volume fraction correlates with *ATP2* mRNA localization.

**Figure supplement 2.** Respiration induced increased protein production.

similar to *ATP3* in that it showed a large increase in localization according to volume fraction change upon a shift to respiratory conditions and was sensitive to cycloheximide (*Figure 5C*; *Figure 5—figure supplement 1*). However, the nonconserved subunit, *ATP7*, behaved more like *TOM22* in that it was insensitive to cycloheximide and had a much smaller increase in mitochondrial localization in respiratory conditions than *ATP2* or *ATP3* (*Figure 5D*). We then wanted to more globally explore the connection between sensitivity to translation elongation and changes in gene expression during the metabolic shift from fermentation to respiration when mitochondrial volume fraction dramatically

**Table 1.** mRNAs of conserved ATP synthetase components are localized to mitochondria in respiratory and CHX conditions. ATP complex gene conservation was obtained from published annotation (**Rühle and Leister, 2015**) and we obtained data of CHX-dependent mitochondrial localization from proximity ribosomal profiling data (**Williams et al., 2014**). The CHX-dependent localization data were consistent with our study. Respiratory-specific localization was determined in this study.

| | Complex | Subunit | Gene | Mitochondrial localization | | Respiratory specific localization |
| --- | --- | --- | --- | --- | --- | --- |
| | | | | -CHX | +CHX | |
| Conserved from bacteria to eukaryote | F1 | α | ATP1 | no | yes | yes |
| | | β | ATP2 | no | yes | yes |
| | | γ | ATP3 | no | yes | yes |
| | | OSCP/δ | ATP5 | no | yes | NA |
| | | ε | ATP16 | no | no | NA |
| | F0 | b | ATP4 | no | yes | NA |
| non-conserved | all the others | | | no | no | NA (ATP7: no) |
| *Rühle and Leister, 2015* | | | | *Williams et al., 2014* | | This study |

increases. We focused on Class II mRNAs that were found to be localized to the mitochondria during respiratory conditions independently of Puf3 (**Saint-Georges et al., 2008**) and subdivided these into ATP3-type mRNAs that were cycloheximide-sensitive in localization to mitochondria during fermentative conditions and TIM50-type that were constitutively localized to mitochondria in fermentative conditions (**Williams et al., 2014**). We used previously generated ribosome profiling data on yeast cells under glucose and glycerol conditions (**Couvillion et al., 2016**). From this data, we found that ATP3-type mRNAs had a significant increase in their protein productive capacity versus TIM50-type as they had >2 fold more ribosomes engaged in translation in the shift from glucose to glycerol (**Figure 5E**). The increase in protein production was caused by both a significant increase in translation efficiency and mRNA levels for ATP3-type mRNAs during respiratory conditions (**Figure 5—figure supplement 2A,B**). From an independent proteomics data set under glucose and glycerol conditions, we also found that ATP3-type genes had an increase in protein abundance during respiratory conditions versus TIM50-type genes (**Figure 5—figure supplement 2C**; **Morgenstern et al., 2017**). These results point to translational elongation sensitivity and condition-dependent localization being general strategies to fine-tune gene expression of certain nuclear-encoded mitochondrial genes.

## mRNA localization to the mitochondria drives enhanced protein synthesis

Recently, it has been shown that there is active cytoplasmic translation on the mitochondrial surface in *Drosophila* and that a subset of nuclear-encoded mitochondrial proteins are translationally regulated by the localization of specific RNA-binding proteins to the mitochondrial surface (**Zhang et al., 2016**). We therefore hypothesized that mRNA localization to mitochondria may be a way to drive the coordinated increase in mitochondrial protein production observed in respiratory conditions. An alternative explanation is that increased translation drives more nascent protein production, which increases mRNA localization to the mitochondria. To directly test these two possibilities, we analyzed the effect of driving mRNA localization to mitochondria on protein expression. To accomplish this, we tethered reporter mRNAs to mitochondria by MS2 sequences. We inserted the MCP protein into the C-terminus of Tom20p and Tom70p, two well-characterized proteins on the outer mitochondrial membrane, and analyzed subsequent protein production (**Figure 6A**). We found that tethering TIM50-flag-GFP-12xMS2 and ATP3-flag-GFP-12xMS2 mRNA to the mitochondria was sufficient to upregulate protein production. Surprisingly, protein production was increased independent of the mRNA harboring an MTS, as an mRNA that contained *flag-GFP* with no mitochondrial coding sequences also showed increased protein production (**Figure 6B–C**). We then analyzed whether tethering to the ER might affect protein production by inserting the MCP protein into the C-terminus

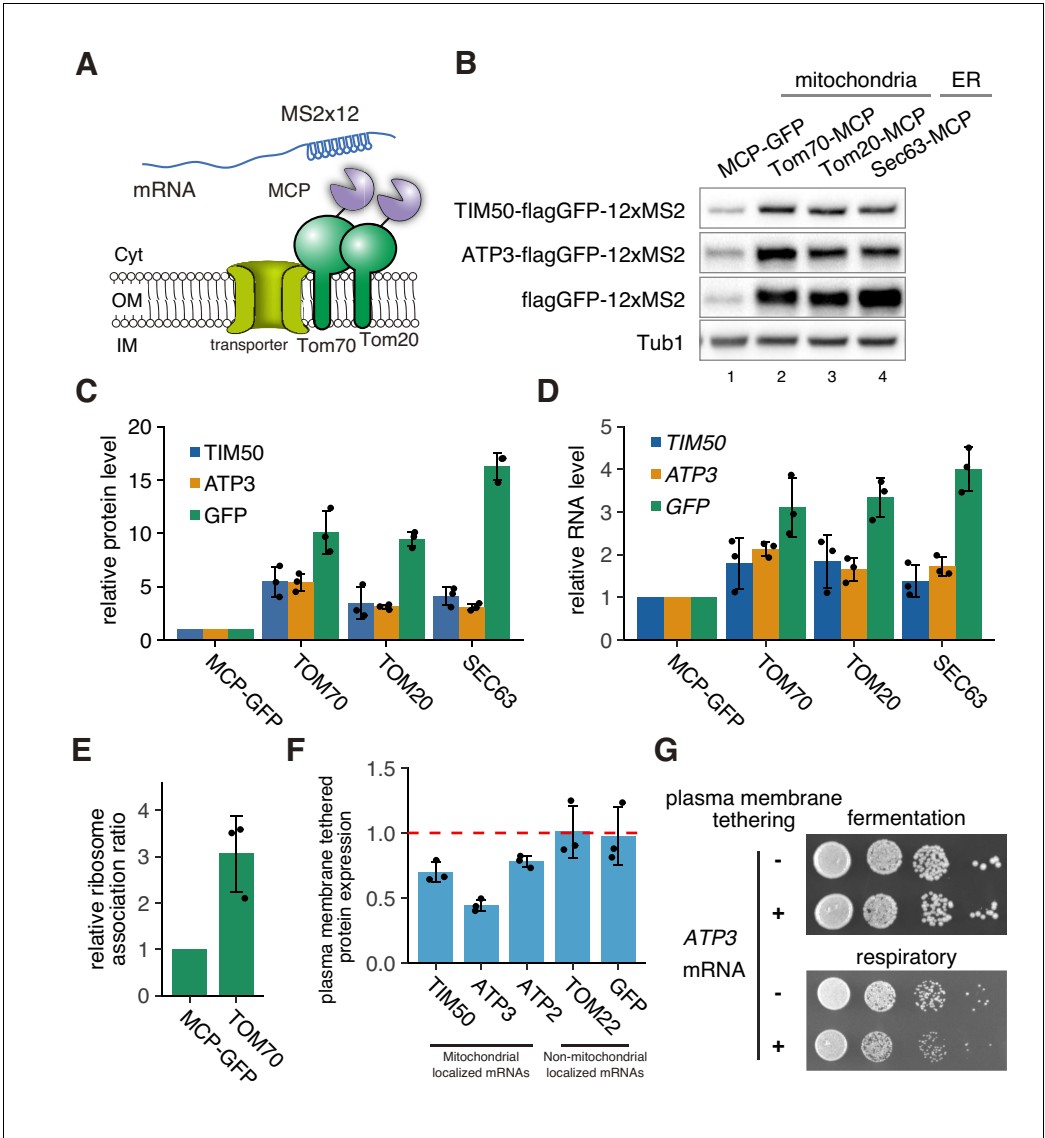

**Figure 6.** mRNA localization to mitochondria enhances its translation. (**A**) Schematic of artificial mRNA tethering to mitochondria using MS2-MCP system. mRNAs harboring a MS2 tandem sequence were tethered to mitochondria through C-terminus MCP tagged Tom70p or Tom20p. (**B**) Protein expression analysis of reporter mRNAs, which are tethered to mitochondria (lanes 2, 3) or ER (lane 4). Protein expression was analyzed using an anti-GFP antibody. Tub1p in the strain harboring flagGFP-12xMS2 reporter genes was used as an internal control. (**C**) Quantification of (**B**). Protein expression was normalized to the No-MCP strains. (**D**) Expression level of reporter mRNAs tethered to mitochondria or ER. mRNA expression was analyzed with RT-qPCR using primers for flag*GFP*. (**E**) Relative ribosome association rate of *flagGFP-12xMS2* expressed with MCP-GFP or TOM70-MCP was calculated by comparison between RNA levels from ribosome-bound and free fractions from sucrose gradient polysome fractionation. RNA levels were quantified via qPCR. (**F**) Protein expression analysis in anchored-away conditions. Protein expression levels in each strain in respiratory conditions were determined by western blotting using anti-flag antibody. Protein expression ratio between strains with and without CaaX is shown as a bar graph. (**C**)-(**F**), Each dot represents an independent experiment. Error bar represents standard deviation of three independent experiments. (**G**) Growth assay for plasma membrane localized mRNA, which consists of an integrated MS2 sequence into the 3'-UTR of genomic DNA. *TIM50* and *ATP3* mRNAs were anchored away to the plasma membrane using CaaX-tag harbored MCP-GFP proteins. Cell growth was tested on YPAD (fermentative) and YPAGE (respiratory) conditions at 30°C for 2 days and 3 days, respectively.

The online version of this article includes the following figure supplement(s) for figure 6:

**Figure supplement 1.** Protein and mRNA expression levels of artificially tethered reporter mRNAs.
**Figure supplement 2.** Ribosome-free and ribosome-bound fraction from Polysome analysis.

*Figure 6 continued*

**Figure supplement 3.** Growth assay for strains with mRNA tethered to either mitochondria, ER, or plasma membrane.

of Sec63p. We also saw increased protein production when mRNA was tethered to the ER, suggesting that the surface of both of these organelles may harbor the capacity for enhanced protein synthesis. In addition to increased protein expression, we also observed increased mRNA levels when mRNAs were tethered to the mitochondria. However, the ratio of protein to mRNA was much higher (*Figure 6D*, *Figure 6—figure supplement 1A*). We further observed a 3-fold increase in ribosome association rate from sucrose density fractionated mRNAs for mRNAs that were tethered to the mitochondria (*Figure 6E*, *Figure 6—figure supplement 2*), suggesting that translational efficiency is increased on the mitochondrial surface. To test whether localization to the mitochondria is necessary for optimal protein production during respiratory conditions, we reduced the localization of endogenous *ATP3* and *TIM50* mRNA to mitochondria by directing those mRNAs to the plasma membrane via insertion of a CaaX-tag to the C termini of MCP-GFP proteins during respiratory conditions (*Yan et al., 2016*). This caused a decrease in protein reporter levels of mitochondrial localized mRNAs, such as *TIM50*, *ATP3*, and *ATP2*, but not in non-mitochondrially localized mRNAs, such as *TOM22* and *GFP* alone (*Figure 6F*, *Figure 6—figure supplement 1B*). We next investigated whether enhancing protein synthesis was essential for optimal cell growth. Cells in which *ATP3* mRNA was anchored to the plasma membrane and away from the mitochondria in respiratory conditions showed a growth defect, whereas ER tethering of mRNAs, which does not impair protein synthesis, did not affect cell growth (*Figure 6G*; *Figure 6—figure supplement 3*). This suggests that localization of mRNA to mitochondria is important for optimal cell growth because it drives enhanced protein synthesis during respiratory conditions.

## Discussion

During fluctuating environmental conditions, cells must be able to control gene expression in order to optimize fitness. We demonstrate that yeast cells can use the geometric constraints that arise from increased mitochondrial volume fraction during respiratory conditions to increase condition-dependent mitochondrial localization for a subset of nuclear-encoded mitochondrial mRNAs. We favor the hypothesis that the geometric constraints of mitochondrial volume fraction are an important factor impacting mRNA localization for two reasons: first, our simple mathematical model incorporating mitochondrial volume fraction and various binding affinities is able to recapitulate the mRNA localization effects we see in cells. Second, the relationship between mitochondrial volume fraction and mRNA localization holds across a multitude of experimental perturbations, which presumably impact mitochondrial function in very different ways. *reg1Δ* changes mitochondrial volume fraction exclusively by increasing mitochondrial volume, whereas chloramphenicol and a nutrient shift from glucose to glycerol media increase mitochondrial volume fraction by both increasing mitochondrial volume and decreasing cytoplasmic volume. While *reg1Δ* and glycerol media increase oxidative phosphorylation, chloramphenicol inhibits mitochondrial translation and respiratory function. Finally, *sch9Δ* has a reduced mitochondrial volume but an increased vacuolar volume. This increase in vacuolar size decreases the accessible cytoplasm, thereby increasing the mitochondrial volume fraction, and increases *ATP3* mRNA localization to the mitochondria. While we believe the geometric constraints of the cell are important for mRNA localization, we cannot completely exclude the possibility that there are secondary factors such as receptor concentrations that also contribute to the mRNA localization, yet as we do not find them to be correlated with mRNA localization across all of our diverse perturbations, we do not believe receptor concentrations are the primary drivers of the mRNA localization changes we observe.

Our results also indicate that translation duration plays an important role in mRNA localization to the mitochondria. We believe that translation duration and mitochondrial volume fraction are interconnected, as both will impact the probability of the mitochondria interacting with a competent mRNA, meaning the MTS is exposed while still attached to the mRNA as a nascent-polypeptide (*Figure 7A*). *ATP3* mRNA, which has fast translation elongation compared to *TIM50*, is in a

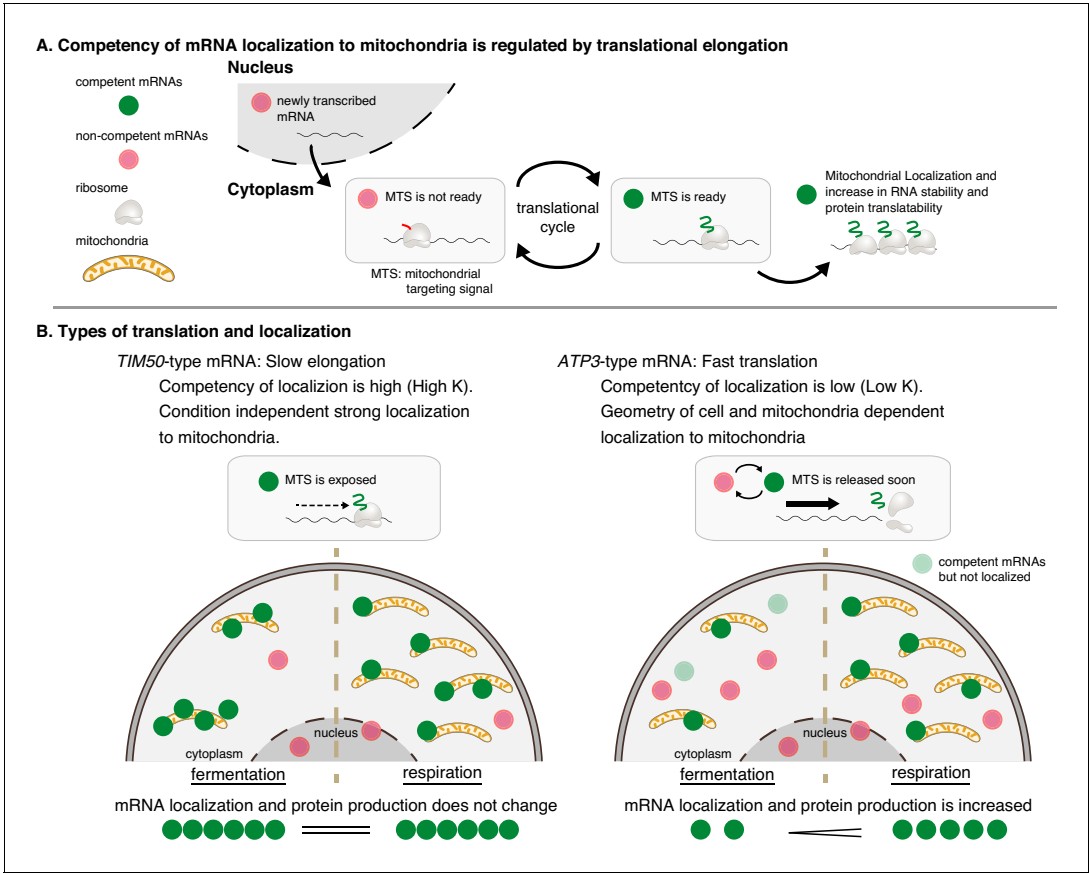

**Figure 7.** Mitochondrial volume fraction controls protein synthesis of nuclear-encoded mitochondrial proteins. (**A**) Competency of mRNA localization to mitochondria is regulated by translational elongation. After the MTS is translated and exposed from the ribosome the mRNA becomes competent for localization to the mitochondria for as long as the mRNA is associated with the nascent chain. Slow elongation will extend the time an mRNA is competent. Mitochondrially localized mRNAs experience an increase in RNA stability and protein translatability. (**B**) Mitochondria can coordinate gene expression during times of metabolic need via mitochondrial-volume-fraction-based control and simple chemical kinetics of nuclear-encoded mRNA localization. *TIM50*-type: mRNAs with high affinity to mitochondria are always associated with mitochondria and thus not much affected by geometrical features. *ATP3*-type: mRNAs with low affinity for mitochondria localization are greatly affected by geometrical features of cells, that is mitochondrial volume fraction. Fast translation elongation leads to quick release of the mRNA/nascent chain complex, and results in a quick return to a non-competent state. When mitochondrial volume fraction is high in respiratory conditions, mRNA localization to mitochondria is increased, and protein synthesis is induced by its localization.

competent state for a shorter period of time, so during fermentative conditions it is less likely to interact with mitochondria while in a competent state (*Figure 7B*). Conversely, ribosome stalling along *TIM50* mRNA prolongs the time the mRNA is in a competent state, and this increases its probability of interacting with mitochondria even when in low mitochondrial volume fraction conditions (*Figure 7B*). As the mitochondrial volume fraction of cells increases as they shift to respiratory conditions, there is a higher probability of competent mRNAs interacting with the mitochondria, even if they are only competent for a short time (*Figure 7B*). While we show that one way to increase translation duration is a translation elongation stall caused by polyproline sequences, other mechanisms that increase translation duration including increasing ORF length, rare codons, and mRNA structures (*Schuller and Green, 2018*) have similar potential to impact mRNA localization.

The localization of ribosomes on the yeast mitochondrial surface has been known since the 1970 s, implicating localized translation. It has been proposed that the functional relevance of this localized translation may be to facilitate co-translational import (*Lesnik et al., 2015*). The necessity of co-translational import is unclear, as in vitro mitochondrial import systems have found it to be important for the import of some proteins but not for others (*Fujiki and Verner, 1991*; *Schatz, 1979*; *Suissa and Schatz, 1982*). Our work points to localization to mitochondria having an alternative function: to upregulate protein synthesis, as we found that mRNA localization to mitochondria was

sufficient to increase protein synthesis, whereas tethering mRNAs away from the mitochondria to the plasma membrane reduced protein synthesis. The decrease in protein levels when *ATP3* mRNA is anchored to the plasma membrane is also associated with a growth defect during respiratory conditions. Yet localization to the mitochondria is not absolutely required for optimal growth; when *ATP3* mRNAs were targeted away from the mitochondria to the ER, there was no growth defect. This may be because, similar to the mitochondria, there is an upregulation of protein levels when mRNAs are targeted to the ER.

Condition-dependent mRNA localization to the mitochondria as a means to control gene expression could be used to tune the protein composition of the mitochondria to the metabolic needs of the cell. Several observations support this hypothesis. For example, the conserved subunits of ATP synthase are all sensitive to the translation elongation rates in the cell, and these subunits showed similar localization regulation patterns (*Table 1*); this suggests that a mechanism may have evolved that coordinates the expression and stoichiometry of vital subunits of this complex. Interestingly it has been shown that nuclear-encoded components of the $F_1$ ATP synthase complex upregulate the translation of the mitochondrially encoded *ATP6* and *ATP8* subunits potentially contributing to the synchronization of these separated genomes (*Rak and Tzagoloff, 2009*). Further supporting the use of condition-dependent mRNA localization in impacting gene expression is global data showing that cycloheximide-sensitive mRNAs (*ATP3*-type) show a more than 2-fold increase in protein synthesis relative to constitutively localized mRNAs (*TIM50*-type) when cells shift from fermentation to respiration (*Figure 5E*). For *ATP3*-type mRNAs, the increase in protein synthesis during respiratory conditions is driven by both increases in translational efficiency and increased mRNA levels (*Figure 1C–E*, *Figure 5—figure supplement 2*). Similarly, the increase in protein levels seen when mRNAs were tethered to the mitochondria was driven by both an increase in mRNA levels and ribosome engagement (*Figure 6C–E*, *Figure 6—figure supplement 1*). As these reporter mRNAs are under identical promoters, we favor the hypothesis that localization to the mitochondria increases mRNA stability. It is unclear whether translational efficiency and mRNA stability are independently affected by localization to the mitochondria or whether they are interrelated, as previous reports have shown that increased translation initiation can lead to protection from mRNA degradation (*Roy and Jacobson, 2013*). As discussed previously, ribosome stalling increases mRNA binding competency, driving mRNA localization to the mitochondria, along with increased translation initiation and mRNA stability. As translation initiation is thought to be the rate-limiting step in translation for most mRNAs, this would explain our somewhat-contradictory results showing that more ribosome stalling, through polyproline insertion, leads to increased protein synthesis.

How mitochondrially localized mRNAs can increase protein synthesis is still an open question. We consider the simplest explanation, that there may be an increased density of ribosomes on the mitochondrial surface, as previously suggested in *Drosophila* oogenesis (*Zhang et al., 2016*). It also might be true that translation initiation factors or mRNA decay factors have altered activities near mitochondria that produce high levels of ATP. An alternative idea is that there are specialized ribosomes, enriched at the mitochondrial surface, that enhance translation (*Xue and Barna, 2012*). Intriguingly, it was recently found that specific ribosomal protein paralogs are necessary for normal mitochondrial function. *Rpl1bΔ* cells were found to be deficient for growth on nonfermentable carbon sources, whereas *rpl1aΔ* had no growth defect in these conditions. Furthermore, *rpl1bΔ* cells were found to have decreased translation of many mitochondrial proteins during respiratory conditions, including the *ATP3*-type mRNAs *ATP1* and *ATP2* (*Segev and Gerst, 2018*). Determining how mRNA localization increases protein synthesis will increase our understanding on how mitochondria are able to control their composition in relation to the metabolic needs of the cell.

While we have found this gene expression control mechanism in yeast, we speculate that higher eukaryotic cells could also use mitochondrial mRNA localization, controlled by translation elongation and mitochondrial volume fraction, to regulate protein synthesis. Hinting at this, a recent paper exploring global mRNA localization in mammalian cells found that many ATP synthase mRNAs had enhanced localization to the mitochondria in a cycloheximide-dependent manner, similar to *ATP3*-type mRNAs in yeast (*Fazal et al., 2019*). Further exploration into the role of mitochondrial volume fraction and translation elongation in mRNA localization to the mitochondria will provide insight into the regulation of mitochondria in health and disease.

## Materials and methods

### Yeast strains and plasmids

The yeast strains and plasmids used are listed in *Supplementary file 1* and the oligonucleotides used for plasmid construction are described in *Supplementary file 2*. To reduce variability among the constructed yeast strains, the strains were created either through integration of a linear PCR product or a plasmid linearized through restriction digest. Yeast strains harboring the genomic MS2 sequence were constructed using pSH47 and pLOXHIS5MS2L, a gift from the J. Gerst laboratory, as previously described (*Gadir et al., 2011*; *Haim-Vilmovsky and Gerst, 2009*). To constitutively visualize mRNAs, pRS405*CYC1p-MS2-4xGFP* (TTP080) was constructed as follows: *MCP-4xGFP* sequence was PCR amplified from pMS2CPGFP(x4) (*Gadir et al., 2011*) and integrated into the pRS405 vector under the *CYC1* promoter. The strains expressing MCP-4xGFP from 2 sets of integrated plasmids were selected through microscopy screening and used for further experiments. To visualize the mitochondria matrix, pRS406GPDp-Su9-mCherry was constructed as follows: Su9 (subunit 9 of the $F_0$ ATPase (Su9(1-69))) sequence was PCR amplified from pvt100-dsRed (*Rafelski et al., 2012*) and the yomCherry sequence was PCR amplified from pFA6a-link-yomCherry-SpHis5 (*Lee et al., 2013*), a gift from the K. Thorn laboratory, and integrated into pRS406 vector under a *GPD* promoter. The strains expressing Su9-mCherry from two sets of integrated plasmids were selected through microscopy screening. To construct plasmids shown in *Figure 3A*, (TTP134 and TTP136), endogenously MS2 sequence-tagged genes were amplified from genomic DNA of strains TTY1160 and TTY439, and inserted into pRS403 vector plasmids (TTP133 and TTP135), respectively. Annealed cytERM (C450_2C1) primers were integrated into TTP133 and TTP135 by combination of Gibson assembly and PCR. To construct TIM50-ΔMTS plasmid (TTP245), MTS region 4–300 bp was swapped from TTP133 by combination of Gibson assembly and PCR. GFP tagging was performed with PCR mediated homologous recombination using pFA6a-link-yoGFP-SpHis5 (*Lee et al., 2013*) and integrations were confirmed by PCR. Deletion mutant strains were constructed with PCR mediated homologous recombination using pFA6a-hphMX6, a gift from the J. Wilhelm laboratory, and integrations were confirmed by PCR. To construct the plasmids used in *Figure 2* and *3*, yoGFP fragment was PCR-amplified from pFA6a-link-yoGFP-SpHis5 (*Lee et al., 2013*) or an iRFP fragment was PCR-amplified from NHB084 with a flag sequence and inserted into TTP133 to construct TTP155 and TTP146, respectively. Synthetic reporters were created by Gibson Assembly of a PCR-amplified *TIM50* (1-300nt), *TIM50* (301-1431nt), *ATP3* (1-300nt) and/or *ATP3* (301-936nt). To construct the polyproline deletion constructs TTP174 and TTP179, deletions were introduced by PCR-based site-directed mutagenesis. To endogenously insert MCP proteins into the C-terminus of Tom70p, Tom20p, Sec63p, we generated TTP119 and TTP153 by replacing a PCR amplified MCP fragment from pMS2CPGFP(x4) (*Gadir et al., 2011*) with yoGFP of pFA6a-link-yoGFP-SpHis5 and pFA6a-link-yoGFP-CaUra3 (*Lee et al., 2013*). MCP tagging was performed with PCR mediated homologous recombination using TTP119 and TTP153. Integrations were confirmed by PCR followed by sequencing. Plasma membrane anchor MCP-CaaX was constructed by PCR-based site-directed mutagenesis using TTP080. The strains expressing MCP-2xGFP-CaaX from two sets of integrated plasmids were selected through microscopy screening and used for further experiments.

### Microscopy

Single molecule mRNA visualization with mitochondria was performed as follows: Yeast cells were grown in YPA medium containing 2% glucose (fermentative) or 3% glycerol + 2% ethanol (respiratory) with 15 ml glass tube at 30˚C with rotator speeds of 60 rpm. 300 µl of mid-log phase wild-type yeast cells (OD600 of 0.4–0.7), grown in appropriate medium, were harvested and placed into an Y04C microfluidic chamber, controlled by the CellASIC Onix system. 300 µl YPAD were placed into the flow-wells, the chambers were loaded with cells at 3 psi, and medium was continuously flowed at 3 psi. Cells were imaged at 30˚C with a Yokogawa CSUX Spinning Disk Confocal (Solamere Technology Group) mounted on a Nikon Eclipse Ti chassis motorized inverted microscope, located at the Department of Developmental and Cell Biology (UCI). Imaging was performed using a 100x/1.49 NA oil APO TIRF objective with the correction collar set manually for each experiment and a 1x tube lens (pixel size 0.084 µm). Z-stacks (300 nm steps) were acquired in the fluorescent channel (33 ms exposure) on a Hamamatsu electron-multiplying charge-coupled device (EMCCD) camera. Imaging

was controlled using MicroManager ImageAcquisition (v1.4.16). For CHX treatment, a flow-well containing a final concentration of 100 µg/ml Cycloheximide (C7698; Sigma-Aldrich) was open and an image was taken every 3 s. To directly analyze the effect of *bona fide* inhibition of translation, we supplemented 1,10-Phenanthroline (131377; Sigma-Aldrich), which inhibits transcription, to CHX with final concentration of 250 µg/ml. For LTM treatment, a flow-well containing a final concentration of 50 µM Lactimidomycin (506291; MilliporeSigma) was open for 20 min and an image was taken every 3 s. Image data for *Figure 2E* and *Figure 2—figure supplement 3* as well as C-terminus integrated GFP intensity data were collected as follows: Mid-log phase wild-type yeast cells (OD600 of 0.4 to 0.7) were grown in appropriate medium and 100 µl were placed into a 96-well Glass Bottom Plate (Cellvis LLC). For chloramphenicol treatment, cells were grown in YPAD medium containing a final concentration of 1 mg/ml chloramphenicol through the experiment (~16 hr) (C0378; Sigma-Aldrich). Cells were imaged at 23°C with 3 s interval by a Perkin Elmer UltraView Vox Spinning Disk Confocal mounted on an Olympus IX81 inverted microscope with Yokogawa CSU-X1-A1 spinning disk head, located at the UCSD School of Medicine Microscopy Core. Imaging was performed using either 100x/1.4 NA or 60x/1.42 NA oil objective with the correction collar set manually for each experiment (pixel size 0.068 µm or 0.111 µm). Z-stacks (300 nm steps) were acquired in the fluorescent channel on an EMCCD Hamamatsu 14 bit 1K × 1K camera. Imaging was controlled using Volocity (PerkinElmer). Image data for *Figure 3—figure supplement 1* were collected as follows: Mid-log phase wild-type yeast cells (OD600 of 0.4 to 0.7) were grown in YPD medium and 100 µl were placed into a 96-well Glass Bottom Plate (Cellvis LLC). Cells were imaged at 23°C with 4 s interval by an Eclipse Ti2-E Spinning Disk Confocal with Yokogawa CSU-X1 (Yokogawa) with 50 µm pinholes, located at the Nikon Imaging Center UCSD. Imaging was performed using SR HP APO TIRF 100 × 1.49 NA oil objective with the correction collar set manually for each experiment (pixel size 0.074 µm). Z-stacks (300 nm steps) were acquired by a Prime 95B sCMOS camera (Photometrics). Imaging was controlled using NIS-Elements software (Nikon).

## Reconstruction of 3D mitochondria and mRNA visualization

To allow accurate visualization of mRNA molecules, multiple MS2 stem-loops are inserted in the 3'-UTR of the mRNA of interest and are recognized by the MCP-GFP fusion protein (*Bertrand et al., 1998*; *Haim-Vilmovsky and Gerst, 2009*). We improved this system by titrating down the MCP-GFP levels until we observed single molecule mRNA foci, which we verified by single molecule RNA FISH (smFISH) (*Hocine et al., 2013*; *Tutucci et al., 2018a*; *Figure 1—figure supplement 1*). We then performed rapid 3D live cell imaging using spinning disk confocal microscopy. We reconstructed and analyzed the spatial relationship between the mRNAs and mitochondria using custom ImageJ plugin Trackmate (*Tinevez et al., 2017*) and MitoGraph V2.0, which we previously developed to reconstruct 3D mitochondria based on matrix marker fluorescent protein intensity (*Rafelski et al., 2012*; *Viana et al., 2015*; *Figure 1A*). We measured the distance between mRNA and mitochondria by finding the closest meshed surface area of the mitochondria matrix. Bias-reduced logistic regression (*Firth, 1992*; *Firth, 1993*) was used to determine which factors influenced the manual tracking of foci in Trackmate. Signal-to-noise ratio (SNR), median intensity of foci, and minimum distance between tracked foci were screened for their contributions to manual tracking of foci. Two-sided p-values were compared for the different variables. The logistic regression analysis shows that median intensity (p-value=0.046) and SNR (p-value=0.036) are the two features detectable by human eyes to track the foci. The code used for the analysis is available from https://github.com/tsuboitat (*Tsuboi and Viana, 2020*; copy archived at https://github.com/elifesciences-publications/Mitograph_Distance).

## Validation of single molecule mRNA of MS2-MCP system

It has been observed that version 4 of the MS2-MCP system is prone to cytoplasmic aggregation, thought to be caused by either stabilization of mRNA or its degradation intermediates by MCP proteins (*Tutucci et al., 2018a*). We improved the MS2-MCP system by titrating down the MCP-GFP levels until we could observe single molecule mRNA foci, which we verified by single molecule RNA FISH (smFISH) (*Hocine et al., 2013*; *Figure 1—figure supplement 1*). Analysing the co-localization rate of candidate mRNA ORF and MS2 sequences using anti-sense probes (*Figure 1—figure supplement 1A*), we confirmed that more than 84% of the ORFs and MS2 sequences are a continuous

stretch in the same mRNA (*Figure 1—figure supplement 1B*). The number of MCP-GFP foci in live cells is statistically no different (Wilcoxon rank-sum test, p-value<0.05, N > 43) than the number of smFISH foci in the tested mRNAs (*Figure 1—figure supplement 1C*). We concluded that the MCP-GFP foci in the live cell images are reliable single molecule mRNAs and not degradation intermediates.

## Single-Molecule FISH and image acquisition and analysis

Single-molecule FISH (smFISH) was performed as previously described (*Tutucci et al., 2018b*). Yeast strains were grown at 23°C in YPAD medium containing 2% glucose and harvested at OD600 0.6–0.8. Cells were fixed by adding paraformaldehyde (15714; 32% solution, EM grade; Electron Microscopy Science) to a final concentration of 4%. Cells walls were removed by resuspension in spheroplast buffer (buffer B containing 20 mM VRC (S1402S; Ribonucleoside–vanadyl complex NEB), and 25U of Lyticase enzyme (L2524; Sigma). Digested cells were seeded on 18 mm polylysine-treated coverslips. The hybridization mix (10% formamide (205821000; ACROS organics), 2x SSC, 1 mg/ml BSA, 10 mM VRC, 5 mM NaHPO4 pH 7.5, 1 mg/µL *E. coli* tRNA, 1 mg/µl ssDNA) was prepared with probe mix (final concentration 125 nM). Cells were then hybridized at 37°C for 4 hr in the dark. Nuclei were stained with 0.5 µg/mL DAPI in 1x PBS. Coverslips were mounted on glass slides using ProLong Gold Antifade Mountant (ThermoFisher). Images were acquired using an Olympus BX61 wide-field epifluorescence microscope with a 100x/1.35 NA UPlanApo objective. Samples were visualized using an X-Cite 120 PC lamp (EXFO) and the ORCA-R2 Digital CCD camera (Hamamatsu). Metamorph software (Molecular Devices) was used for acquisition. Z-sections were acquired at 200 nm intervals over an optical range of 8.0 µm. Image pixel size: XY, 64.5 nm. smFISH probes for *TIM50*, *ATP3* and *TOM22* probes were designed using and purchased from the Stellaris Probe Designer by LGC Biosearch Technologies. MS2 sequence probes, gifted from the R. Singer laboratory, were synthetized by Invitrogen-Thermo Fisher and labelled in the lab using Cy3 dyes (Amersham) as previously described (*Tutucci et al., 2018b*). Quantification of the FISH spots in the cytoplasm was performed using a custom ImageJ pipeline and manually detected. If the focus in different channels was within two pixels, it was considered to be co-localized; otherwise, it was considered to be two non-overlapping and distinct foci.

## Definition of 'localization' and 'association' to mitochondria

To analyze the association of mRNA to mitochondria, we first defined the 'localized' threshold as twice the size of the mode (190 nm) of the distance between *TIM50* mRNA and mitochondria, which were treated with translation elongation and transcription inhibitors cycloheximide (CHX) and 1,10-Phenanthroline (PHE), respectively, expecting that the majority of *TIM50* mRNAs associate with the mitochondrial surface in these conditions (*Figure 1—figure supplement 2*). We further classified mRNA localization to reflect a stable or transient association using this 0.19 um threshold. Since the translational elongation rate is 9.5aa/sec (*Shah et al., 2013*), which implies that it takes more than 30 s to translate reporter genes (more than 300aa), we defined a mitochondria-associated mRNA as being co-localized to the mitochondria for at least 3 s (two consecutive time points).

## Segmentation of cell boundaries and analysis of cell volume

We segmented cell boundaries from the GFP field of the image, which we acquired for mRNA localization analysis in *Figure 1B*, using the ImageJ plug-in Trainable Weka Segmentation (*Arganda-Carreras et al., 2017*). Cell volume was calculated through this boundary. To analyze the effective mitochondrial volume fraction for analyzing mutant strains, we segmented the non-cytoplasmic area as vacuole. To acquire the volume fraction of the vacuole, we chose the center focal plane of each cell in bright field image and segmented the outer cell shape and inner black area, which we assumed as non-cytoplasmic area. Then, fit the ellipse to both outer cell shape and inner non-cytoplasmic area and calculate the volume using shorter radios of the fitted ellipse as depth.

## Single molecule mRNA *in silico* experiment and mathematical modeling

We used Paraview (http://www.paraview.org) to create 3D meshes for cell and mitochondria boundaries based on the segmentations. To model ideal Brownian particles in the cytoplasm, we generated particle trajectories using a random walk (diffusion coefficients (*D*) = 0.1 µm$^2$/s) (*Wu et al.,*

*2016*) in the space between the cell wall and mitochondria surface for at least 40,000 trajectories per cell. Then the proportion of mitochondrial localization per cell (as we defined previously) was quantified. Because the particles were generated as space filling between the cell wall and mitochondrial surface and the mitochondria is a tubular structure, the proportion of mitochondrial localization $R_{localized}$ was linearly correlated with mitochondria volume fraction as described in the following equation (*Figure 2—figure supplement 2A*, black line).

$$R_{localized} \propto \frac{mitochondrial\ surface\ area}{cell\ volume} \propto mitochondrial\ volume\ fraction$$

We assessed whether the localization behavior for *TOM22* mRNA, which we assumed to be a non-mitochondrial localized mRNA, matched with that of a Brownian particle. Although the proportion of mitochondrial localization for *TOM22* mRNA was linearly correlated with the mitochondrial volume fraction, the slope was different from that of a Brownian particle. We hypothesized this was because we had not considered the other cell compartments, such as nucleus (7% of cell volume) (*Jorgensen et al., 2007*), vacuoles (10% of cell volume) (*Chan et al., 2016*), and mitochondria itself (10% of cell volume; this study). We determined that the difference in this proportion was caused by this organelle volume and to account for it, we minimized the cell size accordingly. To fit the *TOM22* mRNA localization proportion line to the Brownian particles we set the effective cell volume to 66%. This was below our expectation, but we found this acceptable since we had not included other organelles such as ER or lipid droplets in the virtual cell to study localization.

We assumed that mitochondrial localized mRNA concentration [mRNA$_{localized}$], free diffusing mRNA concentration [mRNA$_{free}$], and the concentration of mitochondrial surface point where mRNA can bind [mito$_{surface}$] are in thermodynamical equilibrium. We then hypothesized that some mRNAs may associate with the mitochondrial surface by a sequence-specific association (e.g. via an MTS) with a constant K such that:

$$\left[mRNA_{free}\right]\left[mito_{surface}\right] \rightleftharpoons \left[mRNA_{localized}\right]$$

$$K = \frac{\left[mRNA_{localized}\right]}{\left[mito_{surface}\right]\left[mRNA_{free}\right]}$$

*K* is also thermodynamically given by

$$K = e^{\frac{-\Delta G_{localization}}{k_B T}}$$

where $k_B$ is Boltzmann's constant, T is temperature, and $\Delta G_{localization}$ is the free energy gain from mitochondrial localization. For the Brownian particle-like *TOM22*, we assumed a $\Delta G_{localization}$ value of 0; therefore, TOM22 had the lowest equilibrium constant, which we called $K_0$. We then hypothesized that any arbitrary mRNA has an equilibrium constant of $K = AK_0$, where A thus represents a fold-change in affinity between that mRNA and mitochondria compared to a freely diffusing mRNA. As the value of A increased in the simulation, the proportion of mitochondrial localization of the mRNA for a given mitochondrial volume fraction increased as well (*Figure 2—figure supplement 2B*). The proportion of mitochondrial localization $R'_{localized}$ for an arbitrary mRNA when $K = AK_0$ is fitted to the following equation using the linear trend line for proportion of localization $R_{localized_0}$ of *TOM22* mRNA. Upon the same mitochondrial volume fraction, we acquire $R'_{localized}$ as

$$R'_{localized} = A \cdot K_0 \left[mito_{surface}\right]\left[1 - R'_{localized}\right]$$

$$R'_{localized} = A \frac{R_{localized_0}}{\left[1 - R_{localized_0}\right]}\left[1 - R'_{localized}\right]$$

$$R'_{localized} = \frac{A \cdot R_{localized_0}}{1 + (A-1)R_{localized_0}}$$

Thus, we show that the probability of an mRNA localizing to a mitochondrion is dependent upon both the mitochondrial concentration and the free energy change of localizing.

### Quantification of protein expression in each cell from image data

Quantification of GFP-tagged mitochondrially-localized protein expression was performed using a custom analysis pipeline. The MitoGraph was applied to the mCherry channel to identify the locations of mitochondria and the resulting binary mask was then segmented to determine projections of individual mitochondria. Next, the sum of the intensity of the GFP channel covered by binary mask from each cell was measured using a custom ImageJ script.

### Western blotting

Yeast cells were grown in YPA medium containing 2% glucose or 3% glycerol + 2% ethanol. The cells were harvested when the culture reached an OD600 of 0.8. The protein products of various GFP reporter genes were detected by western blotting using an anti-GFP antibody (11814460001; Roche) and a Mouse IgG (H+L) Secondary Antibody (32430; Thermo Fisher). For the detection of the flag-tagged proteins, anti-flag antibody (F1804; Sigma-Aldrich) was used. A mouse anti-a-tubulin antibody (12G10, Developmental Studies Hybridoma Bank) was used as a loading control. The intensities of the bands on the blots were quantified using Molecular Imager ChemiDoc XRS+ System (Bio-Rad) and ImageJ. The relative levels of the protein products were determined by comparison with a standard curve prepared using a series of dilutions.

### Polysome analysis

Yeast cells were grown exponentially at 30°C and harvested by centrifugation. Cell extracts were prepared with mortar and pestle as described previously (*Tsuboi et al., 2012*). The equivalent of 10 A260 units was then layered onto linear 10% to 50% (w/w) sucrose density gradients. Sucrose gradients (10%–50% sucrose in 20 mM HEPES [pH 7.4], 5 mM magnesium chloride, 100 mM potassium chloride, 2 mM DTT, 100 µg/ml CHX) were prepared in 14 × 89 mm polyallomer tubes (Beckman Coulter) with a gradient master. Crude extracts were layered on top of the sucrose gradients and were centrifuged at 30,000 rpm in a SW-28 rotor for 3 hr at 4°C. Gradients were fractionated using a gradient fractionator and UA-6 detector (ISCO/BRANDEL). RNA from ribosome-free and ribosome-bound fraction (monosome – polysome) as shown in *Figure 6—figure supplement 2* were purified from each fraction using guanidine hydrochloride and processed for RT-qPCR.

### RT-qPCR

RNA was extracted using the MasterPure Yeast RNA Purification Kit (Epicentre). cDNA was prepared using ProtoScript II Reverse Transcriptase (NEB #M0368X) with a combination of oligo(dT) primers and random hexamers according to the manufacturer's instructions. mRNA abundance was determined by qPCR using SYBR Green PCR Master Mix (Applied Biosystems) and primers specific for *FLAG-GFP* (ATGGACTACAAGGACGACG and CCTCACCACTCACAGAAAAC), *OM14* (TCACCAC-CACAACAATAAGAAG and AGCGTCAAAAGACCCAGAG), *OM45* (ACAGTAATCCTTTGAAACGCC and TCACCCCATCCTTCTAAACC), and *ACT1* (GAGAGGCGAGTTTGGTTTCA and TCACCCGGCC TCTATTTTC). The mRNA levels were normalized to *ACT1* abundance, and the fold change between samples was calculated by a standard ΔΔCt analysis.

### Data and materials availability

The code used for analyzing the distance between mRNA and mitochondrial surface and generating random walk trajectory is available from https://github.com/tsuboitat. Further information and requests for resources, scripts, and reagents should be directed to and will be fulfilled by the lead contact, T.T. (ttsuboi@ucsd.edu).

## Acknowledgements

We thank members of the Zid laboratory as well as T Endo, S Iwasaki, G Goshima, E Koslover, W Wang and V Bilanchone for helpful discussions and A Subramaniam and S Mukherji for feedback on the paper. Receipt of the MS2-tagging plasmids and yeast-optimized fluorophore plasmids from Dr. J Gerst and Dr. K Thorn is gratefully acknowledged. We thank M Zid, A Guzikowski and V Harjono for critically reading the manuscript. This work was supported in part by National Institutes of Health GM57071 (to RHS), startup funds from UCI, NSF grant MCB-1330451 and Ellison Medical

Foundation (to SMR), startup funds from UCSD and from the National Institutes of Health R35GM128798 (to BMZ), and NINDS P30NS047101 (to UCSD microscopy Core). ET was supported by Swiss National Science Foundation Fellowships P2GEP3_155692 and P300PA_164717. TT acknowledges support from a Japan Society for the Promotion of Science (JSPS) for a research abroad fellowship and postdoctoral fellowship (18J00995), and Uehara Memorial Foundation for research abroad fellowship.

## Additional information

### Funding

| Funder | Grant reference number | Author |
|---|---|---|
| Japan Society for the Promotion of Science | Research abroad fellowship | Tatsuhisa Tsuboi |
| Japan Society for the Promotion of Science | 18J00995 | Tatsuhisa Tsuboi |
| Uehara Memorial Foundation | Research abroad fellowship | Tatsuhisa Tsuboi |
| National Institute of General Medical Sciences | R35GM128798 | Brian M Zid |
| National Science Foundation | MCB-1330451 | Susanne M Rafelski |
| Ellison Medical Foundation | New Scholar Award In Aging | Susanne M Rafelski |
| National Institute of General Medical Sciences | GM57071 | Robert H Singer |
| Swiss National Science Foundation | Fellowship P2GEP3_155692 | Evelina Tutucci |
| Swiss National Science Foundation | Fellowship P300PA_164717 | Evelina Tutucci |

The funders had no role in study design, data collection and interpretation, or the decision to submit the work for publication.

### Author contributions

Tatsuhisa Tsuboi, Conceptualization, Resources, Data curation, Software, Formal analysis, Fundingacquisition, Validation, Investigation, Visualization, Methodology, Writing - original draft, Writing - review and editing; Matheus P Viana, Software, Methodology, Writing - review and editing; Fan Xu, Formal analysis, Investigation; Jingwen Yu, Software, Formal analysis, Investigation, Methodology; Raghav Chanchani, Data curation, Validation, Methodology; Ximena G Arceo, Writing - original draft; Evelina Tutucci, Joonhyuk Choi, Methodology; Yang S Chen, Formal analysis, Methodology; Robert H Singer, Resources, Funding acquisition, Methodology; Susanne M Rafelski, Conceptualization, Resources, Software, Supervision, Funding acquisition, Methodology, Project administration, Writing - review and editing; Brian M Zid, Conceptualization, Resources, Formal analysis, Supervision, Funding acquisition, Investigation, Writing - original draft, Writing - review and editing, Project administration

### Author ORCIDs

Tatsuhisa Tsuboi (iD) https://orcid.org/0000-0003-3249-030X
Evelina Tutucci (iD) http://orcid.org/0000-0002-1998-7146
Robert H Singer (iD) http://orcid.org/0000-0002-6725-0093
Brian M Zid (iD) https://orcid.org/0000-0003-1876-2479

### Decision letter and Author response

Decision letter https://doi.org/10.7554/eLife.57814.sa1
Author response https://doi.org/10.7554/eLife.57814.sa2

## Additional files

### Supplementary files
- Supplementary file 1. Yeast strains and plasmids used in this study.
- Supplementary file 2. List of oligonucleotides used for plasmid construction.
- Transparent reporting form

### Data availability
All data generated or analysed during this study are included in the manuscript and supporting files.

The following previously published datasets were used:

| Author(s) | Year | Dataset title | Dataset URL | Database and Identifier |
|---|---|---|---|---|
| Couvillion MT, Soto IC, Shipkovenska G, Churchman LS | 2016 | Synchronized translation programs across cellular compartments | https://www.ncbi.nlm.nih.gov/geo/query/acc.cgi?acc=GSE74454 | NCBI Gene Expression Omnibus, GSE74454 |
| Williams CC, Jan CH, Weissman JS | 2014 | Targeting and Plasticity of Mitochondrial Proteins Revealed by Proximity-Specific Ribosome Profiling | https://www.ncbi.nlm.nih.gov/geo/query/acc.cgi?acc=GSE61011 | NCBI Gene Expression Omnibus, GSE61011 |
| Morgenstern M, Stiller SB, Lübbert P, Peikert CD, Dannenmaier S, Drepper F, Weill U, Höß P, Feuerstein R, Gebert M, Bohnert M, van der Laan M, Schuldiner M, Schütze C, Oeljeklaus S, Pfanner N, Wiedemann N, Warscheid B | 2017 | Definition of a high-confidence mitochondrial proteome at quantitative scale_pure versus crude mitochondria | http://proteomecentral.proteomexchange.org/cgi/GetDataset?ID=PXD006151 | ProteomeXchange, PXD006151 |

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
