## [Decision Letter]

**Acceptance summary:**

Your paper is of great interest in providing evidence that localization of yeast mRNAs to mitochondria increases their translational efficiencies, and that the increased localization of certain mRNAs during a shift from fermentative to respiratory growth is driven by two factors: (i) the increased proportion of cell volume made up by mitochondria in respiratory growth, and (ii) for certain mRNAs including TIM50, a stretch of consecutive proline codons, known to confer a pause during translation elongation, which they hypothesize is required to increase the time available for association of the mitochondrial targeting sequence in the nascent polypeptide with its receptor on the mitochondrial outer membrane.

**Decision letter after peer review:**

Thank you for submitting your article "Mitochondrial volume fraction and translation speed impact mRNA localization to the mitochondria and protein synthesis" for consideration by *eLife*. Your article has been reviewed by three peer reviewers, one of whom is a member of our Board of Reviewing Editors, and the evaluation has been overseen by James Manley as the Senior Editor. The reviewers have opted to remain anonymous.

The reviewers have discussed the reviews with one another and the Reviewing Editor has drafted this decision to help you prepare a revised submission.

Summary:

This paper provides evidence that localization of yeast mRNAs to mitochondria (mito) increases their translational efficiencies, and that the increased mito-localization seen for certain nucleus-encoded mito protein mRNAs during a shift from fermentative (ferm) to respiratory (resp) growth is driven by two factors. One is the increased proportion of cell volume made up by mito in respiratory growth, while the second, at least for *TIM50* mRNA under ferm conditions when the mito volume fraction is low, is a stretch of consecutive proline codons, known to confer a pause during translation elongation, which they hypothesize is required to increase the time available for association of the mito targeting sequence (MTS) in the nascent polypeptide with the MTS receptor on the mito outer membrane and thereby enhance mRNA localization to mito-associated ribosomes. *ATP3* appears to lack an elongation-stalling sequence as inserting the polyproline tract into its mRNA can enhance its mito-association in ferm conditions, which is normally lower than that seen for *TIM50* that naturally contains the PolyPro. This model can also explain the previous observation that inhibiting translation elongation with CHX can enhance mito-association of *ATP3* and certain other mito mRNAs, but not of TIM50, presumably by substituting for translation elongation stalling sequences in those former mRNAs. Consistent with their model, inhibiting initiation vs. elongation with lactimidomycin did not enhance mito. localization as this would prevent synthesis of the MTS. They also did tethering experiments and showed that artificially tethering any reporter mRNA tested (even GFP) to mitochondria increases its translation and mRNA abundance, whereas tethering *TIM50* or *ATP3* , but not *TOM22* or GFP, to the plasma membrane (PM) reduces their translation relative to the untethered state.

The following are the major comments that would likely require additional experimental evidence to be addressed in a satisfactory way:

1) New experiments are needed to measure the levels of the TOM complex, or other relevant complexes (OM14 or SAM-Mdm10), to see if they increase under resp. growth conditions, which if so, could be an important factor beyond mito. volume per se in increasing localization of mito. mRNAs.

2) Present additional imaging data (videos) to convince us that the same mRNAs are being examined in different z-stacks.

3) Provide evidence for increased translation initiation, beyond just showing increased mRNA localization, in response to manipulations that increase the mitochondria volume fraction.

4) Show that deleting the MTS, rather than replacing it with an ER-targeting sequence, impairs mito. localization.

5) Add analysis of control constructs to confirm that the sequence encoding Polyproline increases mito. localization owing to translation elongation through polyproline codons rather than an inhibitory effect of the RNA sequences per se, involving synonomous codon replacements encoding PolyPro or a single base pair insertion to place the polyPro tract out of frame.

6) Provide a more comprehensive analysis of mRNA localization/translation in both fermentative and resp. conditions.

In addition, the paper should be revised, possibly including the title, to acknowledge that the present evidence is inadequate to conclude that mitochondrial volume is a major driver of mito protein localization, and to note that other intrinsic features of the translated ORFs (e.g. MTS, codon usage, ORF length), changes in translation and the level of the constituents involved in tethering under respiratory conditions, and alterations in cellular architecture (e.g. organelle size), may be equally or more important. The paper should also be revised to address as many as possible of the other major comments not touched on above

Reviewer #1:

This paper is interesting in providing evidence that localization of yeast mRNAs to mitochondria (mito) increases their translational efficiencies, and that the increased mito-localization seen for certain nucleus-encoded mito protein mRNAs during a shift from fermentative (ferm) to respiratory (resp) growth is driven by two factors. One is the increased proportion of cell volume made up by mito in respiratory growth, and although this point is established only by correlation, they did a thorough job of modulating the mito volume proportion in different ways and found coherent effects on mito localization of mRNAs. The second important element for mito localization, at least for *TIM50* mRNA under ferm conditions when the mito volume fraction is low, appears to be a stretch of consecutive proline codons, known to confer a pause during translation elongation, which they hypothesize is required to increase the time available for association of the mito targeting sequence (MTS) in the nascent polypeptide with the MTS receptor on the mito outer membrane and thereby enhance mRNA localization to mito-associated ribosomes. *ATP3* appears to lack an elongation-stalling sequence as inserting the polyproline tract into its mRNA can enhance its mito-association in ferm conditions, which is normally lower than that seen for *TIM50* that naturally contains the PolyPro. This model can also explain the previous observation that inhibiting translation elongation with CHX can enhance mito-association of *ATP3* and certain other mito mRNAs, but not of TIM50, presumably by substituting for translation elongation stalling sequences in those former mRNAs. Consistent with their model, inhibiting initiation vs. elongation with lactimidomycin did not enhance mito. localization as this would prevent synthesis of the MTS. They also did tethering experiments and showed that artificially tethering any reporter mRNA tested (even GFP) to mitochondria increases its translation and mRNA abundance, whereas tethering *TIM50* or *ATP3* , but not *TOM22* or GFP, to the plasma membrane (PM) reduces their translation relative to the untethered state.

As listed below, there are a number of important experiments omitted and controls lacking. They have failed to establish clearly that the MTS of *TIM50* is required for mito association of the mRNA, showing only that substituting it with an ER-localization sequence reduces the mito localization. They also need to show that simply deleting the MTS impairs mito localization of *TIM50* mRNA, if this has not already been published, in order to firmly establish that the MTS is actually necessary for mito localization of TIM50. Additional control experiments are needed regarding the polyproline tract to distinguish between an effect of the mRNA sequence on localization independently of translation, versus the putative effect of Pro codons in stalling elongation: rather than just deleting it from *TIM50*, they should also have made synonymous codon replacements by mutating the wobble position of each Pro codon. And in addition to just inserting it in frame in *ATP3*, they should have made a single base, out-of-frame insertion in which the in-frame codons would no longer encode PolyPro. Finally, they should show that the PolyPro tract won't work if it is located too close to the MTS, as the elongation pause needs to be enacted only after the complete MTS has emerged from the ribosome exit tunnel.

There were also a number of cases where they did not examine the effects of manipulating *TIM50* or *ATP3* under resp conditions in addition to ferm conditions (or at least I couldn't tell if both conditions were examined) making it unclear whether the elongation pause is needed for efficient localization only under ferm growth when the mito cell volume is relatively low,

– Ratio of association is a poor identifier of measurements, such as in Figure 1B, as ratio of what to what is not defined. If it's mitochondrial-associated to total mRNA, proportion would be a better term. Also, the language in figure legends can sometimes be misconstrued to indicate a ratio of association between fermentative and respiratory growth; and so more precision is needed in all of the figures where this term is used.

– As noted above, the authors failed to show that the MTS of *TIM50* is actually necessary for mito localization, only that its replacement with an ER-targeting signal reduces mito association. They need a deletion of the MTS versus replacement to make the point convincingly.

– Figure 3D: The legend claims that both ferm and resp conditions were examined, but it appears from the text that these data are only in ferm conditions, unless "ratio" of association means something different than above. In any event, the data in both conditions should be provided.

– Figure 4C-E: the same difficulty as for Figure 3D exists here, of unclear labeling and vague descriptions in the legends. Results from both ferm and resp conditions are needed to determine if the effects of polyproline on localization are restricted to only ferm conditions when mito volume is low.

– The experiments on the polyproline stretch need controls of synonymous codon changes vs. deletion for TIM50; and insertion of Pro codons out of frame for *ATP3*, to distinguish effects of mRNA sequence vs elongation stalling on localization.

– In the tethering experiments of Figure 6, one would like to know whether the effect on tethering of *ATP3* is less in resp vs. ferm conditions; and it's not clear which condition was actually analyzed. They also need to add the control of measuring native *ATP3* and *TIM50* expression in the strain with Tom70-MCP, as their expression should be unaltered.

– It's unclear why PM-tethering reduces translation of mRNAs that are normally mito-localized, but not non-localized mRNAs, as the findings with *TOM22* and GFP mRNAs with MS2 insertions show clearly that there are abundant ribosomes near the PM for translation of these mRNAs, and all of the mRNAs targeted to mito show increased translation. Presumably there are other unknown factors involved allowing *TIM50* and *ATP3* to be translated better when they are tethered to mitochondria vs the PM, but the authors don't consider this issue and should comment on it.

Reviewer #2:

The gene expression control events highlighted by this work are interesting and speak to the finely tuned nature of a cell to respond to cellular and environmental change. The data is of high quality and well presented. I also find the data to be thought provoking, but I am not convinced that mRNA localization is directly linked to mitochondrial volume fraction vs. binding interactions driven by the nascent chain and difference in translation across metabolic programs. Overall, I am supportive of publication if the points raised below can be adequately addressed.

1) How does mitochondrial composition change with organelle growth? It is not addressed how the TOM complex, or other relevant complexes (OM14 or SAM-Mdm10), change with mitochondrial volume or respiratory activity. For instance, if the density of these complexes are maintained as the membrane grows, or even increase with respiration, this would mean an increase in the total number of binding sites on the mitochondrial surface. As import of MTS proteins and insertion of *Tom22* involves one or more of these complexes, it may be changes in receptor numbers that drive association of mRNAs through the nascent polypeptide chain, not volume per se.

2) Is ORF length a driver of mitochondrial localization? Of the mRNAs studied, the proteins encoded are 476 (*TIM50*), 311 (*ATP3*), and 152 (*TOM22*) residues in length. Yet, the authors do not address the correlation of length with the percent mRNA localized. This question is furthered by the observation that among the ATP complex subunits, the shortest ORF (*ATP16*, 160 residues) is the one that is not impacted by CHX. Williams et al. 2014 has noted that there was correlation between ORF length and localization, with the majority of nascent chains observed at mitochondria being >180 residues in length. A potentially informative experiment to address this issue would involve assaying mRNA localization and protein expression levels in *TIM50*/*ATP3*/*TOM22*-GPF reporters in response to fermentative vs. respiratory growth conditions.

3) The authors perform mRNA imaging over a time series and define a mRNA as localized if it is within a defined distance of the mitochondria in two consecutive imaging frames ~3 seconds apart. Mitochondria can move, an mRNA could release and rebind between z-stacks, or different mRNAs could bind sites physically close together at different times. As such, how is it that the authors know they are counting the same mRNA in consecutive z-stacks? This should be discussed and example images/videos provided to detail how this issue is dealt with. Assuming there is not much change between time points, the authors should be able to provide binding durations for each mRNA. I would expect such dwell time information to be valuable in terms of the modeling data in Figure 2 and could be used to favor or possibly exclude models explaining how/why different mRNAs are localized to the mitochondria surface (e.g. co-translational import, engagement with mitochondrial associated ribosomes, or transient interactions with translational/RNA-binding proteins needed for translation).

Reviewer #3:

In this manuscript, Tsuboi et al. investigated the mRNAs localization of nuclear encoded mitochondrial proteins in yeast under fermentative and respiratory growth conditions. They used single mRNA imaging with MS2 system in live cells with fluorescently labeled mitochondria. The authors observed that three types of behaviors: constitutively (*TIM50*), partially (*ATP3*) and not localized (*TOM22*). They found that *ATP3* localization to mitochondria increases when yeast was changed from fermentative to respiratory condition, while *TIM50* and *TOM22* have less changes. The mRNA localization was correlated to mitochondrial volume fraction. The localization depends on the MTS downstream coding sequence and the authors claim that the localization increased translation initiation. By mutation and translation inhibition, the authors found out that the decreased translation elongation localizes mRNA to mitochondria. Finally, by comparing mis-localized RNAs, the authors found out decreased protein production and some growth defects. The authors proposed a model that mitochondrial RNA localization increases protein translation.

Here the authors used single molecule approach to measure the distance between mRNA and the mitochondria directly, which is technically impressive and nice. Overall, the reviewer think that the data basically agrees with the previous transcriptome wide RNA localization data and the model proposed there: translation drives RNA localization on to the mitochondria, the degree of localization and the CHX sensitivity depend on the relative position of MTS and the downstream sequences in the mRNA. However, I think the data and the proposed model has some gaps, that requires tuning down the claim or further experimental confirmation.

1) The authors concentrated on MTS driven mitochondrial mRNA localization. As proposed before, it mainly depends on the location of MTS within the ORF and the downstream sequence (the length and the plausible stalling sequence). The authors observed that the association with mitochondria of certain mRNAs (*ATP3*) increased when yeast shifted from fermentative to respiratory condition. The authors claim that mitochondria volume fraction "regulate" the fraction of association. I think this is a chicken and egg question. The translation efficiencies of mRNAs may also explain these observations. *ATP3* mRNAs are not actively translated in fermentation condition. The translation efficiency dramatically increases (translation regulation) when the growth condition changed to respiration, which in turn drives the localization of the mRNAs to mitochondria. It makes sense that when fraction of mitochondria is increased, there is a higher chance for MTS to meet the receptor on it, which results in higher colocalization. The authors used ways to manipulate mitochondria fraction and indeed observed more interaction. But in those experiment, there is no direct evidence that increased localization results more translation initiation.

2) Mathematical modeling: the authors claim that *ATP3* and *TIM50* mRNAs has certain affinity to mitochondria. The authors claim the affinity to be "strength of mRNA-specific association". But if the localization is mainly due to translation and MTS binding to the receptor, what does these binding affinities mean? Should it just be the function of translation efficiencies of mRNAs? Should the translation of mRNA explain all the localization? If the model can differentiate whether localization of mRNA increases translation efficiency, it would be more convincing and exciting.

3) The authors claimed increased translation initiation based on western, GFP fluorescence. Those final protein products may also depend on the protein stability. For example, if *TIM50* or *ATP3* are not synthesized on the surface of mitochondria, the nascent proteins may have lower stability and get degraded, because they may require co-transaltional translocation. The decreased protein can also result in growth phenotype.

4) The authors used combination of endogenous labeling and reporters. Sometimes, it is confusing which strains were used because it was not clearly stated. For example, MCP-GFP and *ATP3*-flag-GFP cannot be in the same strain, right? First, it cannot be used to image RNA. Second, the qPCR measurement of mRNA level using GFP primer include both MCP-GFP and *ATP3*-flag-GFP (Figure 6). So are there MCP-RFP used to visualize RNAs?

---

## [Author Response]

Revisions for this paper:The following are the major comments that would likely require additional experimental evidence to be addressed in a satisfactory way:1) New experiments are needed to measure the levels of the TOM complex, or other relevant complexes (OM14 or SAM-Mdm10), to see if they increase under resp. growth conditions, which if so, could be an important factor beyond mito. volume per se in increasing localization of mito. mRNAs.

Firstly, we want to emphasize that the mitochondrial volume fraction, not mitochondrial volume is important for mRNA localization in terms of space restriction/ cellular constraints as shown for *TOM22* mRNA localization and simulation results (Figure 2B, 2D, Figure 2—figure supplement 2). From our modeling we are able to recapitulate general trends in our experimental data by using mRNAs with constant affinities across conditions and just adjusting the spatial restrictions from increasing mitochondrial volume fraction. While our simulation doesn’t completely recapitulate our experimental data, and this could be partially explained by changes in relevant receptors, we think it is beyond the scope of this paper to completely characterize all aspects of mRNA localization in this single manuscript. To touch on this issue though we explored as suggested the expression of the TOM complex, and other relevant complexes (OM14 or SAM-Mdm10) in fermentative versus respiratory conditions using data from Morgenstern et al., 2017. This dataset shows a 3.5 fold increase in *ATP3* and no change in *TIM50* when they compare protein expression levels in glucose and glycerol which are very similar to our results (Figure 1D, E). From their experiments, they showed protein expression are changed as follows.

The protein expression of TOM and most other complexes did not change appreciably in glucose and glycerol, except for OM14 and OM45. As we have perturbations that change mitochondrial volume fraction and *ATP3* localization independent of nutrients we sought to explore if OM14 and OM45 change in these perturbations. We were able to recapitulate the large changes in OM14 and OM45 RNA levels by RT-qPCR in respiratory conditions, but we see a slight decrease of OM14 and no significant change in OM45 in *sch9*∆ and *reg1*∆ strains, where we see a significant increase in mitochondrial volume fraction and *ATP3* mRNA localization. This doesn’t mean OM14 and OM45 play no role in localization during respiratory conditions but that mitochondrial volume fraction is a key factor impacting mRNA localization. We believe further exploration of this relationship is beyond the scope of this manuscript.

We have added the following to the Results section:

“An alternative possibility is that expression of the import machinery or the NAC receptor correlate with mitochondrial volume fraction and that these are what is important for the changes in *ATP3* mRNA localization. […] While we are able to recapitulate the large increase in expression of *OM14* and *OM45* in respiratory conditions, we do not find a significant increase in the expression of these RNAs in *reg1∆* and *sch9∆* mutant strains (Figure 2—figure supplement 5).”

And to the Discussion section: “While we believe the geometric constraints of the cell are important for mRNA localization, we cannot completely exclude the possibility that there are secondary factors such as receptor concentrations that also contribute to the mRNA localization, yet as we do not find them to be correlated with mRNA localization across all of our diverse perturbations, we do not believe receptor concentrations are the primary drivers of the mRNA localization changes we observe.”

2) Present additional imaging data (videos) to convince us that the same mRNAs are being examined in different z-stacks.

We included videos for *TIM50*, *ATP3* and *TOM22* mRNAs in fermentative conditions as Video 2. We also added videos for *TIM50* and *TOM22* mRNAs with addition of cycloheximide and 1,10-phenanthroline for Figure 1—figure supplement 2 as Video 1.

3) Provide evidence for increased translation initiation, beyond just showing increased mRNA localization, in response to manipulations that increase the mitochondria volume fraction.

We show that mRNAs that are elongation rate sensitive in their localization to the mitochondria have a small (~15%) but significant increase in translation efficiency upon a switch to respiratory conditions when these mRNAs are found to be localized to the mitochondria (Figure 5—figure supplement 2A). For the specific mRNAs (*ATP2* and *ATP3*) that we have directly shown have increased mRNA localization in respiratory versus fermentative conditions, we see a 32% and 42% increase in translation efficiency respectively during respiratory conditions (Source Data 1).

Overall we have toned back the language on the increase in translation initiation, and focused instead on the increase in protein synthesis that accompanies mRNAs being localized to the mitochondria as we feel that both mRNA stability and translation efficiency changes are important for the fine tuning of gene expression driven by mRNA localization.

4) Show that deleting the MTS, rather than replacing it with an ER-targeting sequence, impairs mito. localization.

Because of Covid we did not have access to the original microscopes, so instead we used another microscope and visualized mRNA with 4 second interval. Cells were imaged at 23℃ with an Eclipse Ti2-E Spinning Disk Confocal with Yokogawa CSU-X1 (Yokogawa) with 50 µm pinholes, located at the Nikon Imaging Center UCSD. Imaging was performed using SR HP APO TIRF 100x 1.49 NA oil objective with the correction collar set manually for each experiment (pixel size 0.074µm). Z-stacks (300nm steps) were acquired by a Prime 95B sCMOS camera (Photometrics). Imaging was controlled using NIS-Elements software. We have added this as Figure 3—figure supplement 1.

5) Add analysis of control constructs to confirm that the sequence encoding Polyproline increases mito. localization owing to translation elongation through polyproline codons rather than an inhibitory effect of the RNA sequences per se, involving synonomous codon replacements encoding PolyPro or a single base pair insertion to place the polyPro tract out of frame.

Our lactimidomycin data (Figure 3A) shows that translation initiation is key for the localization of *TIM50* mRNA to the mitochondria, and that upon lactimidomycin treatment *TIM50* and *ATP3* have indistinguishable localization phenotypes even though *TIM50* mRNAs contain the 7 consecutive polyprolines and *ATP3* doesn’t, arguing against a role for the RNA sequence directly having a role in the differential localization between *TIM50* and *ATP3*.

6) Provide a more comprehensive analysis of mRNA localization/translation in both fermentative and resp. conditions.

We added newly analyzed data of proportion of association for the chimeric reporters and *TIM50*-P7∆ reporter genes in respiratory conditions. These data are included as Figure 3—figure supplement 2 and Figure 4D.

In addition, the paper should be revised, possibly including the title, to acknowledge that the present evidence is inadequate to conclude that mitochondrial volume is a major driver of mito protein localization, and to note that other intrinsic features of the translated ORFs (e.g. MTS, codon usage, ORF length), changes in translation and the level of the constituents involved in tethering under respiratory conditions, and alterations in cellular architecture (e.g. organelle size), may be equally or more important. The paper should also be revised to address as many as possible of the other major comments not touched on above

We have changed the title to reflect the more inclusive language of translation duration, as we agree that other factors that impact translation duration can impact localization. We have added the following paragraph to the Discussion “While we show that one way to increase translation duration is a translation elongation stall caused by polyproline sequences, other mechanisms that increase translation duration including increasing ORF length, rare codons, and mRNA structures(Schuller and Green, 2018) have similar potential to impact mRNA localization.”

Reviewer #1:[…]There were also a number of cases where they did not examine the effects of manipulating TIM50 or ATP3 under resp conditions in addition to ferm conditions (or at least I couldn't tell if both conditions were examined) making it unclear whether the elongation pause is needed for efficient localization only under ferm growth when the mito cell volume is relatively low,– Ratio of association is a poor identifier of measurements, such as in Figure 1B, as ratio of what to what is not defined. If it's mitochondrial-associated to total mRNA, proportion would be a better term. Also, the language in figure legends can sometimes be misconstrued to indicate a ratio of association between fermentative and respiratory growth; and so more precision is needed in all of the figures where this term is used.

Thank you for your clarification. We changed the y axis of Figures 1-5 and figure supplements to “proportion of associated RNA”. We also edited the figure legends of them.

– As noted above, the authors failed to show that the MTS of TIM50 is actually necessary for mito localization, only that its replacement with an ER-targeting signal reduces mito association. They need a deletion of the MTS versus replacement to make the point convincingly.

As discussed in major additional experiment 4 above.

– Figure 3D: The legend claims that both ferm and resp conditions were examined, but it appears from the text that these data are only in ferm conditions, unless "ratio" of association means something different than above. In any event, the data in both conditions should be provided.

As discussed in major additional experiment 6 above.

– Figure 4C-E: the same difficulty as for Figure 3D exists here, of unclear labeling and vague descriptions in the legends. Results from both ferm and resp conditions are needed to determine if the effects of polyproline on localization are restricted to only ferm conditions when mito volume is low.

As discussed in major additional experiment 6 above.

– The experiments on the polyproline stretch need controls of synonymous codon changes vs. deletion for TIM50; and insertion of Pro codons out of frame for ATP3, to distinguish effects of mRNA sequence vs elongation stalling on localization.

As discussed in major additional experiment 5 above.

– In the tethering experiments of Figure 6, one would like to know whether the effect on tethering of ATP3 is less in resp vs. ferm conditions; and it's not clear which condition was actually analyzed. They also need to add the control of measuring native ATP3 and TIM50 expression in the strain with Tom70-MCP, as their expression should be unaltered.

As it is written in the legend, we showed the protein expression level in respiratory conditions to see if altered protein expression leads slow growth in respiratory condition. For *TIM50* and *ATP3* we observed similar protein expression differences in fermentative/vegetative conditions, while for *ATP2* there was less of an effect in fermentative conditions. As we were focused on the respiratory conditions where we see the slow growth effect we only included the impact of tethering away on protein production during those conditions.

**Author response image 1. sa2fig1:** Tethering to plasmamembrane reduce protein production in vegetative and fermentative condition. N=3.

– It's unclear why PM-tethering reduces translation of mRNAs that are normally mito-localized, but not non-localized mRNAs, as the findings with TOM22 and GFP mRNAs with MS2 insertions show clearly that there are abundant ribosomes near the PM for translation of these mRNAs, and all of the mRNAs targeted to mito show increased translation. Presumably there are other unknown factors involved allowing TIM50 and ATP3 to be translated better when they are tethered to mitochondria vs the PM, but the authors don't consider this issue and should comment on it.

We believe this is not a *TIM50* or *ATP3* specific issue, and that any mRNA that is localized to the mitochondrial surface has increased protein synthesis as seen for the cytosolic protein *Flag-GFP* in Figure 6B, C. So it is not that the plasma membrane is specifically limiting protein synthesis of mito-mRNAs localized there, but that mRNAs pulled away from the mitochondria don’t get the beneficial effects of being localized to the mitochondria and the associated increased protein synthesis. Similarly when we decrease mRNA localization to the mitochondria by deleting prolines in *TIM50*, we see decreased protein synthesis, not because it is localized to the plasma membrane, but because it is not localized to the mitochondrial surface.

We inserted the following statement in Discussion.

“How mitochondrially localized mRNAs can increase protein synthesis is still an open question. […] Determining how mRNA localization increases protein synthesis will increase our understanding on how mitochondria are able to control their composition in relation to the metabolic needs of the cell.”

“Finally, they should show that the PolyPro tract won't work if it is located too close to the MTS, as the elongation pause needs to be enacted only after the complete MTS has emerged from the ribosome exit tunnel.“

We agree with this assumption that MTS needs to be exposed from the ribosome for binding to mitochondria. We inserted a polyproline tract at 100aa from MTS and still see the increased mRNA localization. The ribosome tunnel size is generally thought as 30aa length and we are uncertain if effects from inserting the polyproline tract closer than 100aat would be from the MTS not being exposed or because the MTS was disrupted as the MTS is assumed to be ~30-100aa, but not always perfectly defined.

Reviewer #2:The gene expression control events highlighted by this work are interesting and speak to the finely tuned nature of a cell to respond to cellular and environmental change. The data is of high quality and well presented. I also find the data to be thought provoking, but I am not convinced that mRNA localization is directly linked to mitochondrial volume fraction vs. binding interactions driven by the nascent chain and difference in translation across metabolic programs. Overall, I am supportive of publication if the points raised below can be adequately addressed.1) How does mitochondrial composition change with organelle growth? It is not addressed how the TOM complex, or other relevant complexes (OM14 or SAM-Mdm10), change with mitochondrial volume or respiratory activity. For instance, if the density of these complexes are maintained as the membrane grows, or even increase with respiration, this would mean an increase in the total number of binding sites on the mitochondrial surface. As import of MTS proteins and insertion of Tom22 involves one or more of these complexes, it may be changes in receptor numbers that drive association of mRNAs through the nascent polypeptide chain, not volume per se.

This is fair point and we assessed change of receptors. Please see major additional experiment 1 above.

2) Is ORF length a driver of mitochondrial localization? Of the mRNAs studied, the proteins encoded are 476 (TIM50), 311 (ATP3), and 152 (TOM22) residues in length. Yet, the authors do not address the correlation of length with the percent mRNA localized. This question is furthered by the observation that among the ATP complex subunits, the shortest ORF (ATP16, 160 residues) is the one that is not impacted by CHX. Williams et al. 2014 has noted that there was correlation between ORF length and localization, with the majority of nascent chains observed at mitochondria being >180 residues in length. A potentially informative experiment to address this issue would involve assaying mRNA localization and protein expression levels in TIM50/ATP3/TOM22-GPF reporters in response to fermentative vs. respiratory growth conditions.

We agree with the reviewers points that length of the mRNA will also potentially impact localization to the mitochondrial surface and have changed the title to “translation duration” to better encompass this point. Yet we have also found that length is not the only factor as we can remove the 7 polyprolines from *TIM50* (a <2% length reduction) and see a large drop in localization, while inserting these short residues in *ATP3* increases localization. We also believe that using fluorescent microscopy to explore the impact of length skew the results.

We tested *TIM50-iRFP*, *ATP3-iRFP* by inserting *iRFP* into C-terminus of those genes. Please see major additional experiment 6 above. We see differences in ratio of mitochondrial association with/without *iRFP*. We observed ratio of association as 0.52 in fermentative and 0.7 in respiratory condition for *TIM50* mRNA and ratio of association as 0.15 in fermentative and 0.5 in respiratory condition for *ATP3* mRNA. When we insert *iRFP*, we observed an increase in the ratio of association for *ATP3-iRFP* (0.27 in fermentative and 0.6 in respiratory condition) yet we see a slight decrease in ratio of association for *TIM50-iRFP* (0.37 in fermentative and 0.64 in respiratory condition) compared to the shorter *TIM50*. The additional *iRFP* length of 948nt which can be at least 10-20nm in vivo (Adivarahan et al., 2018) may affect our quantification of localization because of the thresholding we set for a localized mRNA.

3) The authors perform mRNA imaging over a time series and define a mRNA as localized if it is within a defined distance of the mitochondria in two consecutive imaging frames ~3 seconds apart. Mitochondria can move, an mRNA could release and rebind between z-stacks, or different mRNAs could bind sites physically close together at different times. As such, how is it that the authors know they are counting the same mRNA in consecutive z-stacks? This should be discussed and example images/videos provided to detail how this issue is dealt with. Assuming there is not much change between time points, the authors should be able to provide binding durations for each mRNA. I would expect such dwell time information to be valuable in terms of the modeling data in Figure 2 and could be used to favor or possibly exclude models explaining how/why different mRNAs are localized to the mitochondria surface (e.g. co-translational import, engagement with mitochondrial associated ribosomes, or transient interactions with translational/RNA-binding proteins needed for translation).

Visualizing single molecule mRNA and mitochondria simultaneously in 3D was technically challenging experiment. MS2-GFP signals from mRNAs are fast bleaching and we can visualize them for 30 sec, which we are not able to analyze the binding duration time. As described in Materials and methods, we tracked mRNA manually and tested whether those tracks are equal to nearest neighbor search. We cannot exclude the opportunity that we mistrack the same single molecule mRNA in consecutive time points. As we added in major additional experiment 2 above, you may easily track the single mRNA, which associated to mitochondria by eyes, but for freely diffusing mRNAs we rely on justification by nearest neighbor search. We focused on the analysis of ratio of mitochondrial associated mRNA, which binds to mitochondria for consecutive 2 time points, which we are much more confident on those being the same mRNAs.

Reviewer #3:In this manuscript, Tsuboi et al. investigated the mRNAs localization of nuclear encoded mitochondrial proteins in yeast under fermentative and respiratory growth conditions. They used single mRNA imaging with MS2 system in live cells with fluorescently labeled mitochondria. The authors observed that three types of behaviors: constitutively (TIM50), partially (ATP3) and not localized (TOM22). They found that ATP3 localization to mitochondria increases when yeast was changed from fermentative to respiratory condition, while TIM50 and TOM22 have less changes. The mRNA localization was correlated to mitochondrial volume fraction. The localization depends on the MTS downstream coding sequence and the authors claim that the localization increased translation initiation. By mutation and translation inhibition, the authors found out that the decreased translation elongation localizes mRNA to mitochondria. Finally, by comparing mis-localized RNAs, the authors found out decreased protein production and some growth defects. The authors proposed a model that mitochondrial RNA localization increases protein translation.Here the authors used single molecule approach to measure the distance between mRNA and the mitochondria directly, which is technically impressive and nice. Overall, the reviewer think that the data basically agrees with the previous transcriptome wide RNA localization data and the model proposed there: translation drives RNA localization on to the mitochondria, the degree of localization and the CHX sensitivity depend on the relative position of MTS and the downstream sequences in the mRNA. However, I think the data and the proposed model has some gaps, that requires tuning down the claim or further experimental confirmation.1) The authors concentrated on MTS driven mitochondrial mRNA localization. As proposed before, it mainly depends on the location of MTS within the ORF and the downstream sequence (the length and the plausible stalling sequence). The authors observed that the association with mitochondria of certain mRNAs (ATP3) increased when yeast shifted from fermentative to respiratory condition. The authors claim that mitochondria volume fraction "regulate" the fraction of association. I think this is a chicken and egg question. The translation efficiencies of mRNAs may also explain these observations. ATP3 mRNAs are not actively translated in fermentation condition. The translation efficiency dramatically increases (translation regulation) when the growth condition changed to respiration, which in turn drives the localization of the mRNAs to mitochondria. It makes sense that when fraction of mitochondria is increased, there is a higher chance for MTS to meet the receptor on it, which results in higher colocalization. The authors used ways to manipulate mitochondria fraction and indeed observed more interaction. But in those experiment, there is no direct evidence that increased localization results more translation initiation.

We agree with the reviewer’s chicken and egg proposition, to decouple condition specific translation effects from mRNA localization effects we tested the impact of tethering reporter mRNAs to mitochondria during fermentative conditions. For all mRNAs tested we measured increased protein synthesis. We believe this is in part due to increased mRNA stability, but that doesn’t explain the full extent of increased protein production. So to directly assay whether localization increases ribosome loading we performed polysome profiling and found that there is an increased fraction of mRNA associated with ribosomes when an mRNA is localized to the mitochondria, this suggests that in part mitochondrial localization increases protein synthesis through increased translation initiation (Figure 6A-E).

2) Mathematical modeling: the authors claim that ATP3 and TIM50 mRNAs has certain affinity to mitochondria. The authors claim the affinity to be "strength of mRNA-specific association". But if the localization is mainly due to translation and MTS binding to the receptor, what does these binding affinities mean? Should it just be the function of translation efficiencies of mRNAs? Should the translation of mRNA explain all the localization? If the model can differentiate whether localization of mRNA increases translation efficiency, it would be more convincing and exciting.

As much of the actual localization process and import is still an active area of research with many unknowns we sought to use as simplistic model as possible to explore whether space/mitochondrial volume fraction with constant “affinity” could explain the condition specific localization phenotypes we see. From further research in our paper we agree that for *ATP3* and *TIM50* the time an mRNA is competent (MTS exposed) is probably the major factor impacting “strength of mRNA-specific association”. This competency will be impacted by multiple factors including translation initiation rate (translation efficiency), but also mRNA length and as we show elongation rate.

3) The authors claimed increased translation initiation based on western, GFP fluorescence. Those final protein products may also depend on the protein stability. For example, if TIM50 or ATP3 are not synthesized on the surface of mitochondria, the nascent proteins may have lower stability and get degraded, because they may require co-transaltional translocation. The decreased protein can also result in growth phenotype.

This unique idea that mis-co-translational translocation leads to protein degradation is a very interesting question and we were also interested in how the protein stability can change. We observed that half-life (Protein half-life was analyzed through western blotting. Samples were harvested at 0, 30 and 60 min after the addition of CHX (final 100µg/ml). The half-life was calculated by fitting data in log scale.) of the ATP3-GFP protein which is translated at the plasma membrane and the control half-life was both more than 2 hours, and not significantly different from each other. This is intuitive that 2 hours protein half-life does not much affect protein abundance of yeast cells, in which the cell cycle is 1.5 hours, but gene expression (cell cycle) controls protein abundance. Baum et al., 2019 has mathematically shown that cell cycle is more important than protein half-life for protein abundance when the protein half-life is longer than cell cycle. They stated that “It becomes evident that the influence of cell division increases with increasing half-life of either mRNA (Figure 3A, top row) or mRNA and protein.”

4) The authors used combination of endogenous labeling and reporters. Sometimes, it is confusing which strains were used because it was not clearly stated. For example, MCP-GFP and ATP3-flag-GFP cannot be in the same strain, right? First, it cannot be used to image RNA. Second, the qPCR measurement of mRNA level using GFP primer include both MCP-GFP and ATP3-flag-GFP (Figure 6). So are there MCP-RFP used to visualize RNAs?

For visualizing mRNA movements in Figure 3 and 4, we used iRFP as a reporter genes. For Figure 6 we used MCP-GFP as a control in comparison to mito- and ER- tethering and tested protein levels using GFP-antibody and conduct qPCR using flag sequence as a part of amplicon. GFP-antibody binds to both MCP-GFP and flagGFP reporter gene, however we can distinguish the difference by the size.